# Generation and characterization of cardiac valve endothelial-like cells from human pluripotent stem cells

LinXi Cheng[1,2], MingHui Xie[3], WeiHua Qiao [3], Yu Song[4], YanYong Zhang[1], YingChao Geng[1], WeiLin Xu[5], Lin Wang[4,6], Zheng Wang [4,7], Kai Huang [8], NianGuo Dong[3✉] & YuHua Sun [1,2✉]

The cardiac valvular endothelial cells (VECs) are an ideal cell source that could be used for making the valve organoids. However, few studies have been focused on the derivation of this important cell type. Here we describe a two-step chemically defined xeno-free method for generating VEC-like cells from human pluripotent stem cells (hPSCs). HPSCs were specified to $KDR^+/ISL1^+$ multipotent cardiac progenitors (CPCs), followed by differentiation into valve endothelial-like cells (VELs) via an intermediate endocardial cushion cell (ECC) type. Mechanistically, administration of TGFb1 and BMP4 may specify VEC fate by activating the NOTCH/WNT signaling pathways and previously unidentified targets such as ATF3 and KLF family of transcription factors. When seeded onto the surface of the de-cellularized porcine aortic valve (DCV) matrix scaffolds, hPSC-derived VELs exhibit superior proliferative and clonogenic potential than the primary VECs and human aortic endothelial cells (HAEC). Our results show that hPSC-derived valvular cells could be efficiently generated from hPSCs, which might be used as seed cells for construction of valve organoids or next generation tissue engineered heart valves.

[1] Institute of Hydrobiology, Chinese Academy of Sciences, Wuhan, China. [2] University of Chinese Academy of Sciences, Beijing, China. [3] Department of Cardiovascular Surgery, Union Hospital, Tongji Medical College, Huazhong University of Science and Technology, Wuhan, China. [4] Research Center for Tissue Engineering and Regenerative Medicine, Union Hospital, Tongji Medical College, Huazhong University of Science and Technology, Wuhan, China. [5] Wuhan Textile University, Wuhan, China. [6] Department of Clinical Laboratory, Union Hospital, Tongji Medical College, Huazhong University of Science and Technology, Wuhan, China. [7] Department of Gastrointestinal Surgery, Union Hospital, Tongji Medical College, Huazhong University of Science and Technology, Wuhan, China. [8] Department of Cardiovascular Internal Medicine, Union Hospital, Tongji Medical College, Huazhong University of Science and Technology, Wuhan, China. ✉email: Dongnianguo@hotmail.com; sunyh@ihb.ac.cn

Valve disease is one of the most common cardiac defects[1]. No good medical treatments for the dysfunctional valves are available currently, so surgical replacement with mechanical and bioprosthetic valves is the major option. Unfortunately, the current mechanical and bioprosthetic valves have their limitations due to the requirement of lifelong anti-coagulation, poor durability, and lack of self-growth and self-renewal capacity[2,3]. The iPSC-based valve organoids or next-generation tissue-engineered heart valves may offer a potential remedy to the challenge, which however requires the generation of a large number of genuine valvular cells[4,5].

There are two major types of valvular cells: the valvular endothelial cells (VECs) lining the outer surface of the valve cusps and the valvular interstitial cells (VICs) embedded in a stratified extracellular matrix. HPSCs (including human embryonic stem cells and iPSCs) are known to have the potential for unlimited expansion and differentiation into any cell types in a petri dish[6]. Few studies, however, have been focused on the derivation of valve endothelial-type cells, likely due to incomplete understanding of VEC biology. A growing body of studies support the notion that the endocardial cells are derived from $Isl1^+/Kdr^+$ multipotent progenitors via cardiogenic mesoderm cells expressing $T^+/Mesp1^+$[7–9]. Genetic studies using mice and zebrafish models have suggested that the majority of valvular cells are of endocardial cell origin[10]. At E9.5 of mouse embryos or week 5 of human fetus, a subset of endocardial cells at defined regions of the atrio-ventricular (AV) canal and the outflow tract (OFT) (referred as the endocardial cushion cells/ECCs) are induced to loose the contact with neighboring cells, and undergo an EndoMT (endocardial to mesenchymal transition) to form the valve mesenchymal cells or adopt the VEC fate[11]. However, the exact mechanisms by which the ECCs are induced within the endocardium and VEC identity is established remain unclear.

Previous studies have shown that components of multiple signaling pathways (such as BMP, FGF, WNT, NOTCH, and TGF-β) are expressed in a spatial-temporal manner during endocardial cushion formation and EndoMT[12–18], suggesting that these signaling pathways may play key roles in ECC induction and valvular cell formation. Besides, transcriptional regulation is also important for valvulogenesis[10,19]. For instance, NFATc1 is critically important for the maintenance of VEC identity[11,20]. Consistently, NFATc1[low] cells will undergo EndoMT to form VICs while NFATc1[high] cells remain the endothelial phenotype.

In this work, we reported that using a chemically defined xeno-free method, valve endothelial-like cells can be efficiently derived from hPSCs in 9 days. HPSC-derived VELs exhibit morphological, molecular and functional similarities to that of primary VECs isolated from normal human aortic valves. When hPSC-derived VELs were seeded onto the surface of the de-cellularized porcine aortic valve (DCV) matrix scaffolds, they exhibit superior proliferative and clonogenic potential than the primary VECs and HAEC.

## Results

### Efficient generation of ISL1+cardiac progenitor cells from hPSCs
It is widely accepted that the mammalian embryonic heart is primarily originated from cardiac progenitor cells (CPCs) expressing marker genes such as ISL1, NKX2.5, and KDR[10,21–23]. We and several other groups have previously generated the CPCs from hPSCs[8,24–30]. The common theme of the various differentiation protocols is the activation of WNT and BMP signaling pathways, which mimics the signaling requirement during early specification of heart mesoderm[7,8,31].

It has been shown that there is a greater propensity for isolated ISL1+ cardiac progenitors to give rise to endothelium when NKX2.5 is low or absent[8,23]. We aimed to generate ISL1+ KDR+ NKX2.5[low] CPCs from hPSCs by modifying our previously published protocol[25]. HPSCs were initially treated with BMP4 and WNT agonists (WNT3a or the small molecule CHIR99021) for 3 days, and the expression of T, a primitive streak and early mesodermal marker, was monitored daily (Fig. 1a; Supplementary Fig. 1a). Compared to the treatment with BMP4 or WNT3a only, combined treatment with BMP4 and WNT3a resulted in a quick induction of T, reaching its maximum expression levels at day 1 (Supplementary Fig. 1a). Combined treatment with BMP4 and CHIR led to similar results (Supplementary Fig. 1b). Next, day 1$T^+$ cells were treated with bFGF and BMP4 for 6 days, and the expression of CPC markers such as ISL1 and NKX2.5 was examined. The qRT-PCR results showed that ISL1 and KDR were highly induced, peaking at day 2 post-treatment, while NKX2.5 and TBX5 were barely upregulated (Supplementary Fig. 1c, d). Immunofluorescence (IF) staining of day 3 CPCs showed that a vast majority (>90%) of cells were ISL1 positive while cells displayed negligible expression of NKX2.5 (Fig. 1c, d; Supplementary Fig. 1e). Western blot analysis confirmed that ISL1 and KDR were abundantly expressed in day 3 hPSC-derived CPCs (Fig. 1e). Finally, flow cytometry analysis further revealed that approximately 80% of day 3 CPCs were KDR positive and over 93% of cells were ISL1 positive, indicating that hPSC cardiogenic mesodermal differentiation was of high efficiency (Fig. 1f).

Based on the above results, we concluded that ISL1+ KDR+ CPCs could be efficiently generated from hPSCs in 3 days.

### One-step differentiation of CPCs into VEC-like cells
It is widely accepted that endocardial cells are derived from $Isl1^+/Kdr^+$ multipotent CPCs, and that heart valves originate from a subpopulation of endocardial cells called endocardial cushion cells (ECCs) or valve endocardial cells located at defined regions of the AVC and the OFT[7–9]. Recent studies have shown that the ECCs or valve endocardial cells express marker genes such as NPR3, CDH11, EDN1, GATA5, MEF2C, SOX17, THSD1, EMCN, TGFβ2, NOTCH4, DLL3/4, and JAG1 (designated as ECC genes thereafter)[32–36].

Studies using mice and zebrafish models have provided important clues that signaling pathways, such as the BMP, FGF, TGFβ, and NOTCH pathways, may play key roles in the induction of valve endocardial or ECC fate[8,31,34,37–39]. We, therefore, investigated the effects of modulating the BMP, FGF, TGF-β, and NOTCH signaling on day 3 hPSC-derived CPCs, in the presence of VEGFA (Fig. 2a). VEGF has been shown to be important for endothelial cell differentiation from hPSCs[27,35]. Initially, hPSC-derived CPCs were treated with activators or inhibitors of various signaling pathways for 3 days, and the expression of ECC genes was examined. When FGF/ERK signaling was modulated by treatment with different doses of bFGF and FGF/MEK inhibitors PD0325901/PD98059 (Supplementary Fig. 2a), the expression of ECC genes was barely altered. It has been shown that there is an increased expression of both TGFβ and BMPs in the endocardial cushion cells[15,38], and that the combined treatment with BMP4 and TGFb/ActivinA promotes hPSC differentiation into cardiovascular endothelial cell types[30,40]. We speculated that a crosstalk between BMP and TGF-β signaling pathways might be involved in the induction of ECC fate[41,42]. Although administration of BMP4 or TGFb1 moderately altered the expression of ECC genes, combined treatment with BMP4 and TGFb1 led to a marked upregulation of these genes (Fig. 2b; Supplementary Fig. 2b, c). The Notch signaling pathway is well-known to be implicated in endocardial cushion formation[27,39,43]. Surprisingly, modulating Notch signaling (by the addition of different doses of the Notch ligand DLL4

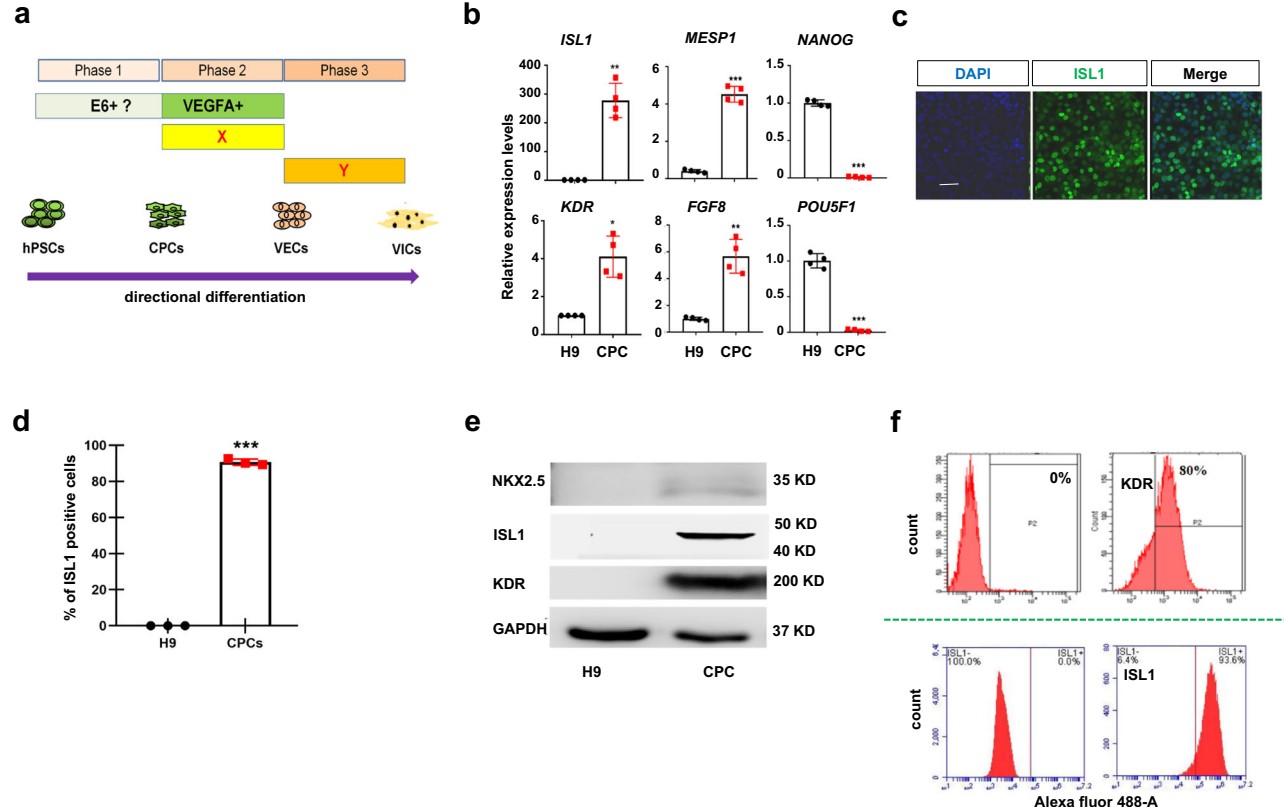

**Fig. 1 PSCs to cardiogenic mesoderm expressing KDR and ISL1. a** Schematic of differentiation hPSCs to CPCs, VEC-like cells, and VIC-like cells. X indicates the future cytokines or small molecules for phase 2; Y indicates the future cytokines or small molecules for phase 3. **b** The qPCR analysis of day 3 hPSC-derived CPCs for the indicated markers. **c** IF staining of day 3 CPCs showing that the majority of cells were ISL1 positive. Scale bar: 50 μm. **d** Quantification analysis of IF staining of (**c**) by ImageJ. Bar graph represents percentage of ISL1 positive cells ± S.D of three independent experiments. **e** WB analysis of day 3 CPCs using the indicated antibodies. **f** Flow cytometry analysis showing the percentage of KDR and ISL1 positive cells, respectively. All experiments were repeated 3 times. The paired *t* test in Graphpad software was used for the statistical analysis. Significant levels are: *$p < 0.05$; **$P < 0.01$; ***$P < 0.001$. Shown are representative data in (**c**, **e**).

or the Notch signaling inhibitor DAPT) had a minimal effect on the expression of ECC genes except *NFATc1* (Supplementary Fig. 2d), and addition of DLL4 in the combined presence of BMP4 and TGFb1 did not further augment ECC gene expression (Supplementary Fig. 2e).

The above data suggested that combined treatment with BMP4 and TGFb1 may promote CPCs to an ECC fate. To further investigate this, day 3 hPSC-derived CPCs were treated with VEGFA/BMP4/TGFb1 for a time course of 9 days, and the dynamic expression of ECC genes was monitored. The treated cells adopted the typical cobber-stone EC morphology from day 2 post-treatment and remained the morphology up to 12 days (Supplementary Fig. 2f). As shown in Fig. 2c, the ECC genes such as *NPR3*, *EMCN*, *MEF2C*, *JAG1*, *DLL4*, and *NOTCH1/4* were quickly induced, most of which exhibited highest expression levels at day 4 post-treatment. CPCs at day 3 post-treatment (or day 6 hPSC-derived cells) were chosen for further analysis as described below. IF staining results showed that day 6 hPSC-derived cells readily expressed general EC markers CD31 and VE-cad, as well as ECC markers NOTCH4, DLL4 and JAG1 (Fig. 2d). Quantitative analysis of IF staining revealed that approximately 90% of the cells were VE-cad/NOTCH4, CD31/JAG1 and VE-cad/DLL4 double positive (Fig. 2e). Flow cytometry analysis of day 6 hPSC-derived cells revealed that approximately 95% and 84% of the cells were JAG1/VE-cad and GATA4/VE-cad double positive, respectively (Fig. 2f; Supplementary Fig. 2g). Thus, day 6 hPSC-derived cells displayed a gene expression profile similar to that of the pre-EMT ECCs[33,35].

With the prolonged treatment with VEGFA/BMP4/TGFb1, the expression of ECC genes gradually decreased. Interestingly, known VEC-specific marker genes such as *NFATc1*, *SMAD6*, and *HAND2* were gradually induced, peaking at day 6 post-treatment (Fig. 2c)[35]. This observation suggested that prolonged treatment with the signaling molecules may lead to the induction of VEC fate. IF staining results showed that day 8 hPSC-derived cells (CPCs at day 5 post-treatment) abundantly expressed VEC markers NFATc1, PROX1, HEY2, and TBX2 (Fig. 2g; Supplementary Fig. 2h). WB analysis confirmed that NFATc1, NOTCH1/3, DLL4, HEY2, and TBX2 were strongly induced in day 8 hPSC-derived cells (Fig. 2h). Finally, flow cytometry analysis revealed that approximately 82% of the cells were NFATc1/VE-cad double positive, and that approximately 67% of cells were TBX2/VE-cad positive (Fig. 2i).

Based on the above results, we concluded that under the combined treatment with VEGFA/BMP4/TGFb1, ECC-like and VEC-like cells could be efficiently derived from hPSCs, in a time dependent fashion.

**HPSC differentiation into VELs recapitulates the development of embryonic VECs.** Next, we asked that to what extent hPSC-derived VELs resembled the genuine VECs, in particular the embryonic VECs. To this end, we consulted and re-analyzed the single-cell RNA-sequencing (scRNA-seq) data of cardiac tissues of human embryos ranging from 5 to 25 weeks of gestation[36]. The VEC cluster can be identified based on the expression of *NFATc1* and

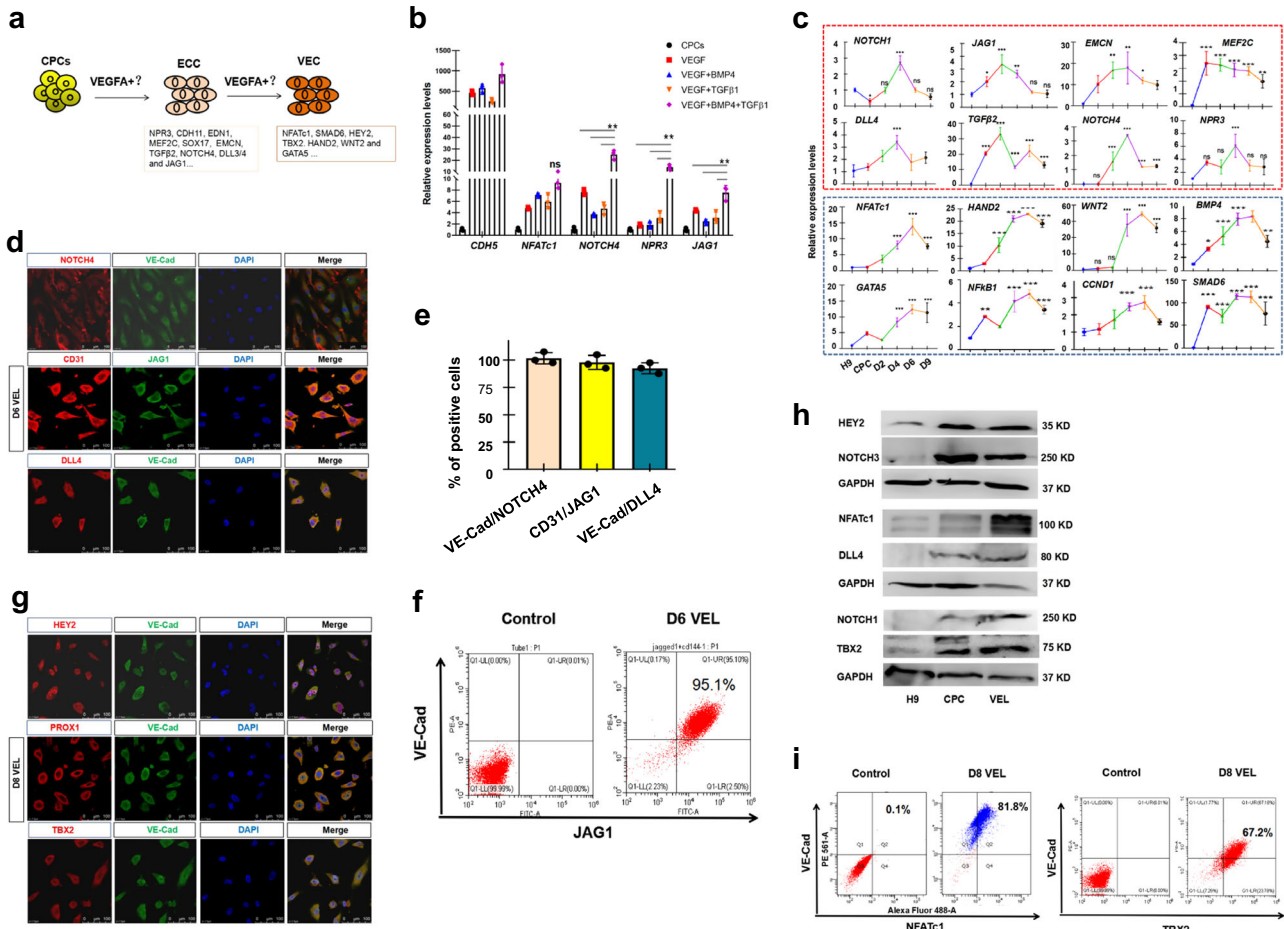

**Fig. 2 One step differentiation of CPCs to VELs. a** Schematic of differentiation hPSC-derived CPCs to ECC and VEC-like cells. **b** The qRT-PCR analysis of the selected ECC genes for day 6 hPSC-derived cells. Note that the combined treatment with VEGF, BMP4, and TGFb1 leads to a marked upregulation of ECC genes. **c** Time-course gene expression analysis of the indicated genes during hPSCs differentiation to CPCs and CPCs to VEC-like cells. The ECC-enriched genes are highlighted by a red dashed box, and the VEC-enriched genes are highlighted by a blue box. **d** IF staining of day 6 hPSC-derived cells, showing double positive cells for NOTCH4/VE-cad, DLL4/VE-cad and CD31/JAG1, respectively. Scale bar: 25 μm. **e** Percentage of NOTCH4/VE-cad-, DLL4/VE-cad- and CD31/JAG1-double-positive cells of (**d**), quantified by ImageJ. Bar graph represents double positive cells ± S.D of three independent experiments. **f** Flow cytometry results showing the percentage of JAG1/VE-cad double positive cells. Left: epitope controls; Right: JAG1/VE-cad antibodies. **g** IF staining of day 8 hPSC-derived VELs, showing double positive cells for HEY2/VE-cad, PROX1/VE-cad and TBX2/VE-cad, respectively. Scale bar: 25 μm. **h** WB analysis for hPSC-derived CPCs and day 8 hPSC-derived VELs, with the indicated antibodies. **i** Flow cytometry results showing the percentage of NFATc1/VE-cad and TBX2/VE-cad double positive cells. The paired *t* test in Graphpad software was used for the statistical analysis. Significant levels are: *$p < 0.05$; **$P < 0.01$; ***$P < 0.001$. Shown are representative data in (**d**, **g**).

*NTRK2*, and the endocardium cluster can be identified based on the expression of *NPR3* and *CDH11* (Fig. 3a; Supplementary Fig. 3a, b). A pseudotime trajectory was built to reveal the developmental progression of embryonic VECs, which showed that embryonic VECs were primarily originated from endocardium of early stages (Fig. 3b). Next, temporal gene expression dynamics were analyzed during the progressive formation of embryonic VECs. We analyzed 2347 genes that are dynamically expressed along the pseudotime trajectory, and identified two major gene groups: G1 with 1104 genes and G2 with 896 genes (Fig. 3c). *Dll4* and *WNT2* are shown as representatives for G1 and G2 genes, respectively (Supplementary Fig. 3c). GO (gene ontology) analysis of G1 genes showed that they were enriched for terms such as cardiovascular development, anatomical structure and multicellular organism development, and regulation of developmental progress. GO analysis of G2 genes revealed an enrichment for terms such as response to organic substance and chemical stimulus, response to cytokine and shear stress, and cytokine-mediated pathways (Supplementary Fig. 3d). KEGG pathway analysis revealed that the G1 genes were enriched for regulation of cardiac development

and NOTCH signaling pathway while the G2 genes were enriched for MAPK, PI3K/AKT and RAS signaling pathways (Supplementary Fig. 3e).

The G1 genes were primarily expressed in endocardial cells of week 5–15 human embryos (early stage), but not in VECs of week 21–25 human embryos (late stage) (Fig. 3c; Supplementary Fig. 3f). The G1 was enriched for cardiac developmental genes such as *FOXC1*, CDH5, *KDR*, *F8*, *vWF*, *FLRT2*, *FN1*, *NKX2.5*, *GATA4*, *PDGFRB*, *WT1*, *ETV5*, and *TBX5*. Remarkably, ECC genes such as *NPR3*, *KDR*, *vWF*, *ENG*, *SOX17*, *EMCN*, *MEF2C*, *JAG1*, *DLL3/4*, *NOTCH4*, and *TGFβ2* belonged to G1. The G2 genes were barely expressed in early stage, but highly expressed in VECs of week 21–25 human embryos (late stage), including *NFATc1*, *NTRK2*, *CDH13*, *HEY2*, *FOS*, *NR4A2*, *CD44*, *CDKN1A*, *CCND1*, *NFKB1*, *CCDC141*, *HEY2*, *HAND2*, *WNT2/4*, *PROX1*, *DKK2/3*, *BMP4*, *ID1/2*, *SMAD6*, *FOXO1*, *GATA2/5*, *THBD*, *MSX1*, *SELE*, *ATF3*, *KLF2/4/9/11*, and *CAV1*. Of which, *NFATc1*, *NTRK2*, *PROX1*, *SMAD6*, *MSX1*, *SELE*, and *IGFBP7* are known VEC-specific or enriched genes. The results strongly indicated

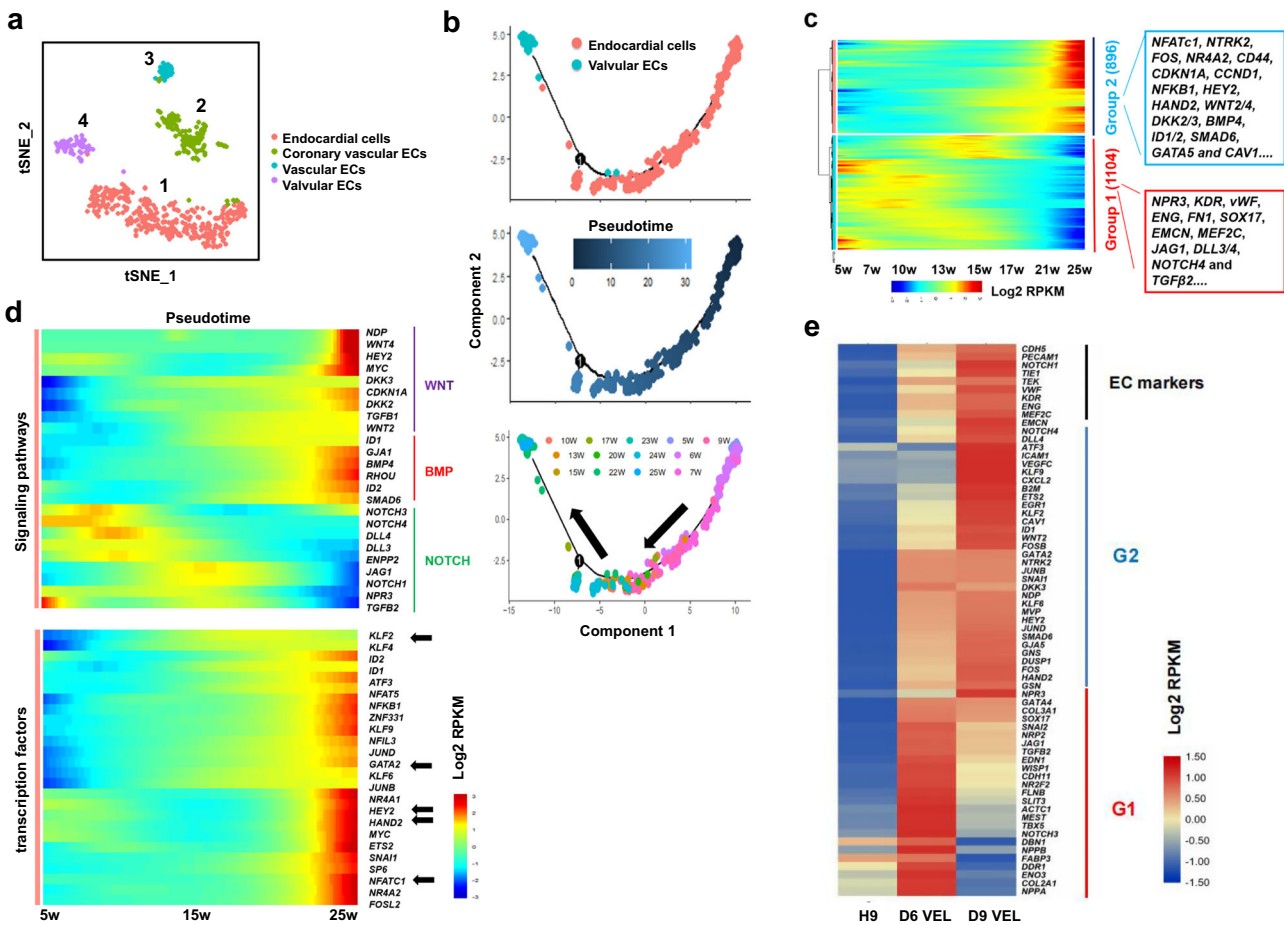

**Fig. 3 Analysis of the primary VECs at single cell level showing that hPSC VEL differentiation recapitulates embryonic VEC development. a** The t-distributed stochastic neighbor embedding (t-SNE) showing unbiased clustering results of endothelial cells[36]. Four different ECs were identified: 1 (endocardial cells), 2 (coronary vascular ECs), 3 (vascular ECs) and 4 (valvular ECs). **b** Pseudotime trajectory showing the distribution of cardiac endothelial cells. Top: distribution of endocardial cells and valvular ECs along pseudotime trajectory; middle: pseudotiming of endocardial and valvular ECs; and bottom: distribution of cells with exact developmental time points. Black arrows show the direction of pseudotime across trajectories. **c** Heat map illustrating the dynamic gene expression patterns of group 1 (G1) and 2 (G2) genes. Note: G1 genes are primarily expressed at early stages; while G2 are highly expressed at later stages. **d** Heat map illustrating the dynamic gene expression patterns of the indicated signaling pathway genes (top panel) and transcription regulatory genes (bottom panel) identified in G2. Black arrows pointing to TFs with well-known function in VEC induction and/or formation. **e** Heat map showing expression of a panel of G1 and G2 genes for day 6 and day 9 hPSC-derived VEC-like cells.

that G1 genes were enriched for ECC markers and G2 genes were enriched for VEC markers.

The dynamic signaling requirement and transcription program for the induction of VEC fate in vivo were characterized over the trajectories. The heat map showed that VEC formation correlated with the upregulation of a panel of TFs, including *NFATc1/5*, *HEY2*, *GATA2*, *ATF3*, *NR4A1/2*, *KLFs*, and *HAND2* (Fig. 3d, bottom)[44,45]. Expression of *KLFs*, *EGR1*, *CXCL2*, and genes reflecting NFkB and JNK activation (*NFKB1*, *JUN*, *FOS*, *FOSB*, and *ATF3*), was in line with the physiological conditions of VECs (shaped by laminar shear flow and fluid shear stress). Based on the expression of signaling components (Fig. 3d, top), NOTCH signaling was activated at early stage, followed by WNT and BMP signaling. Co-expression of a subset of TFs (*NFATc1/HAND2/HEY2*) with WNT- and BMP-related genes strongly suggested that embryonic VEC induction may require both WNT/BMP signaling activation and expression of defined TFs. This was in line with that either the NOTCH/WNT/BMP axis or sustained *NFATc1/HEY2/HAND2* expression is required for normal valvulogenesis[11,43,45].

The identification and characterization of ECC-enriched gene network (G1 genes) and VEC-enriched gene network (G2 genes)

allowed us to better compare our hPSC-derived VELs with embryonic VECs. On one hand, during CPC differentiation into VELs, ECC-enriched genes such as *NPR3*, *KDR*, *EMCN*, *MEF2C*, *JAG1*, *DLL3/4*, *NOTCH1/4*, and *TGFβ2* were firstly induced, peaking at day 4 post-treatment, and VEC-enriched genes such as *NFATc1*, *HAND2*, and *GATA5* were upregulated at later time points, reaching their highest expression levels at day 6 post-treatment (Fig. 2c). This was in line with the heat map analysis of selected G1 and G2 genes: day 6 hPSC-derived cells highly expressed G1 genes and day 9 hPSC-derived VELs abundantly expressed G2 genes (Fig. 3e). Thus, day 6–7 hPSC-derived cells exhibited a gene expression profile analogous to that of ECCs, and day 8–9 hPSC-derived cells displayed a gene expression signature similar to the embryonic VECs. We concluded that the developmental hierarchy of hPSC VEC differentiation faithfully recapitulates embryonic VEC formation in vivo.

**Transcriptome comparison between hPSC-derived VELs and the genuine VECs.** To further investigate the similarity between hPSC-derived VELs and the genuine VECs, bulk RNA-sequencing analysis was performed for hPSC-derived VELs and the primary VECs isolated from normal aortic valves of different

ages (Table 1, Supplementary Table 1). Human ESC H9 cell line (WA09 hESCs), Human foreskin fibroblast cells (HFF), human valvular interstitial cells (hVICs), human umbilical vein endothelial cells (HUVEC) and aortic endothelial cells (HAEC) were

used as controls. An average of 16,000 genes in the primary hVECs were identified, which was similar to that of HUVEC (Supplementary Fig. 4a). Heat map analysis of normalized RNA-seq data showed that day 7 hPSC-derived VELs were globally more similar to the primary VECs than HUVEC and HAEC (Supplementary Fig. 4b). When the heap map analysis was performed using a subset of G2 genes, a similar result was obtained (Supplementary Fig. 4c). We confirmed this by qRT-PCR analysis (Supplementary Fig. 4d, e).

A total of 2,361 highly-expressed genes (FPKM > 10) were shared by day 7 hPSC-derived VELs and the primary hVECs of 9-year-old. GO analysis of the shared genes showed that they were highly related to terms such as cardiovascular and valve development (Supplementary Fig. 4f). KEGG pathway enrichment analysis revealed that TGF-β, Notch, and BMP signaling pathways were highly represented (Fig. 4a). Next, the top 100 genes were subjected to heat map analysis (Fig. 4b). They included a subset of known VEC genes such as *NFATc1*, *NRTK2*, *SMAD6*, *HAND2*, *WNT2*, *GATA5*, *CCND1*, *DKK2/3*, *NFKB1*, *HEY2*, and *ID1/2*, as

**Table 1 The primary VECs isolated from normal human aortic valves.**

| Age (year-old) | Sample numbers | Aortic side VEC (VEC-A) (Yes/No) | Ventricular side VEC (VEC-V) (Yes/No) | VICs (Yes/No) |
|---|---|---|---|---|
| 7 | 1 | Yes | Yes | Yes |
| 9 | 1 | Yes | Yes | Yes |
| 19 | 1 | No | Yes | Yes |
| 20 | 1 | Yes | Yes | Yes |
| 30 | 2 | Yes | Yes | Yes |
| 43 | 2 | Yes | Yes | Yes |
| 50 | 2 | Yes | Yes | Yes |
| 51 | 1 | Yes | Yes | Yes |
| 59 | 1 | Yes | Yes | Yes |

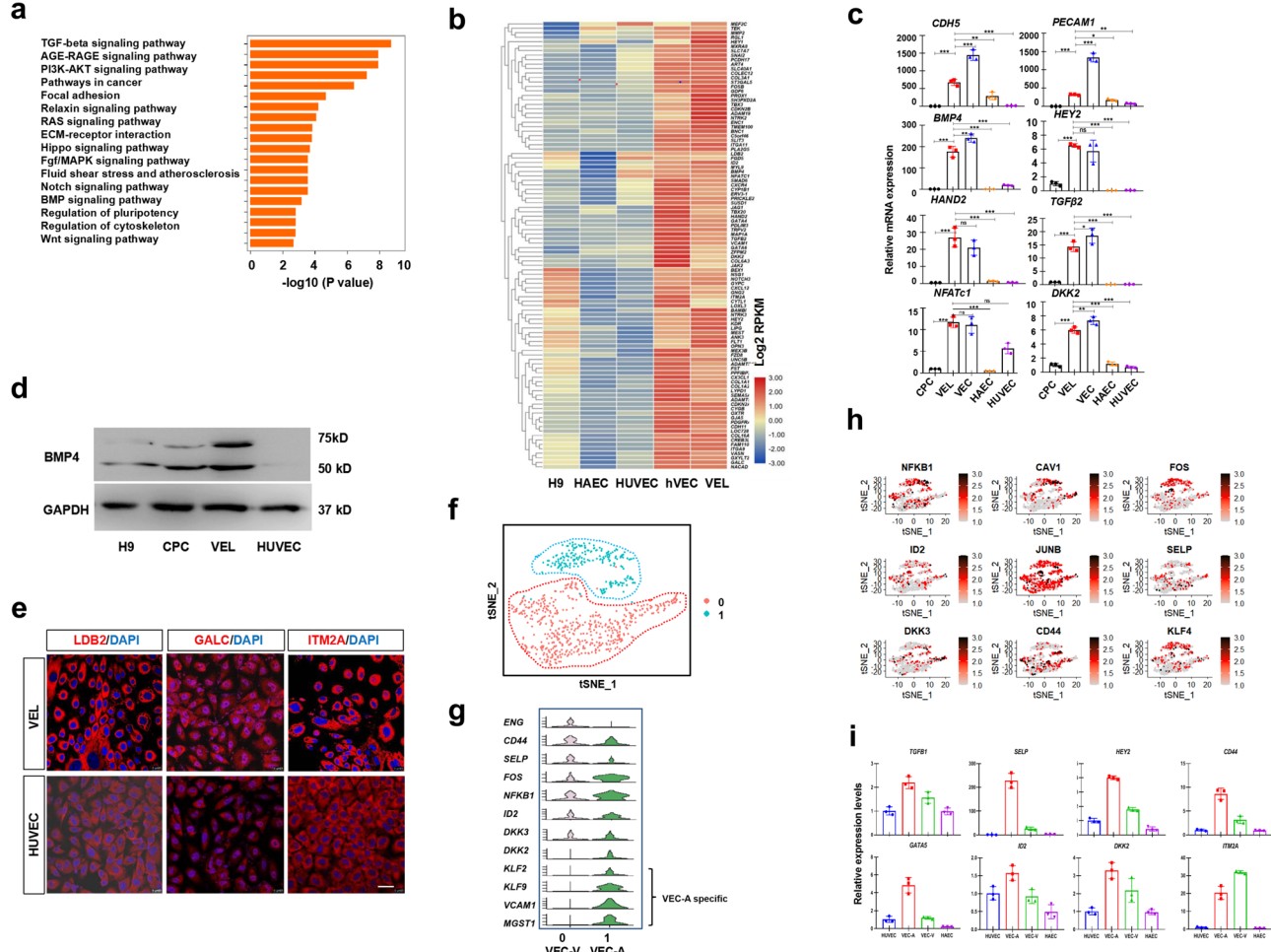

**Fig. 4 Transcriptome comparison between the VELs and the primary VECs. a** KEGG analysis of the shared genes between the VELs and the primary VECs, showing the enrichment of TGF-β, WNT, BMP, and NOTCH signaling pathways. **b** Heat map analysis showing that the majority of top 100 genes were highly expressed in the primary VECs and hPSC-derived VELs, but lowly in HAEC/HUVEC. **c** Validation of the indicated genes by qRT-PCR analysis. **d** WB results showing that BMP4 was expressed at much higher levels in day 7 hPSC-derived VELs than HUVEC. **e** IF staining results showing that the indicated markers were expressed at higher levels in day 7 hPSC-derived VELs than in HUVEC. Scale bar: 100 μm. **f** The t-SNE map showing the second-level clustering of VECs into two subpopulations, designated as 0 and 1. **g** Violin plots showing the expression of indicated genes for subcluster 0 and 1, suggesting that subcluster 0 may represent ventricular side specific VECs and subcluster 1 may represent aortic side specific VECs. **h** tSNE maps showing that the indicated marker genes were indeed highly expressed in the VEC subclusters. **i** qRT-PCR results showing that the indicated genes were expressed much higher in VECs than HAEC and HUVEC. VEC-A and VEC-V represent the aortic and the ventricular side of VECs, respectively. The paired *t* test in Graphpad software was used for the statistical analysis. Significant levels are: *$p < 0.05$; **$P < 0.01$; ***$P < 0.001$.

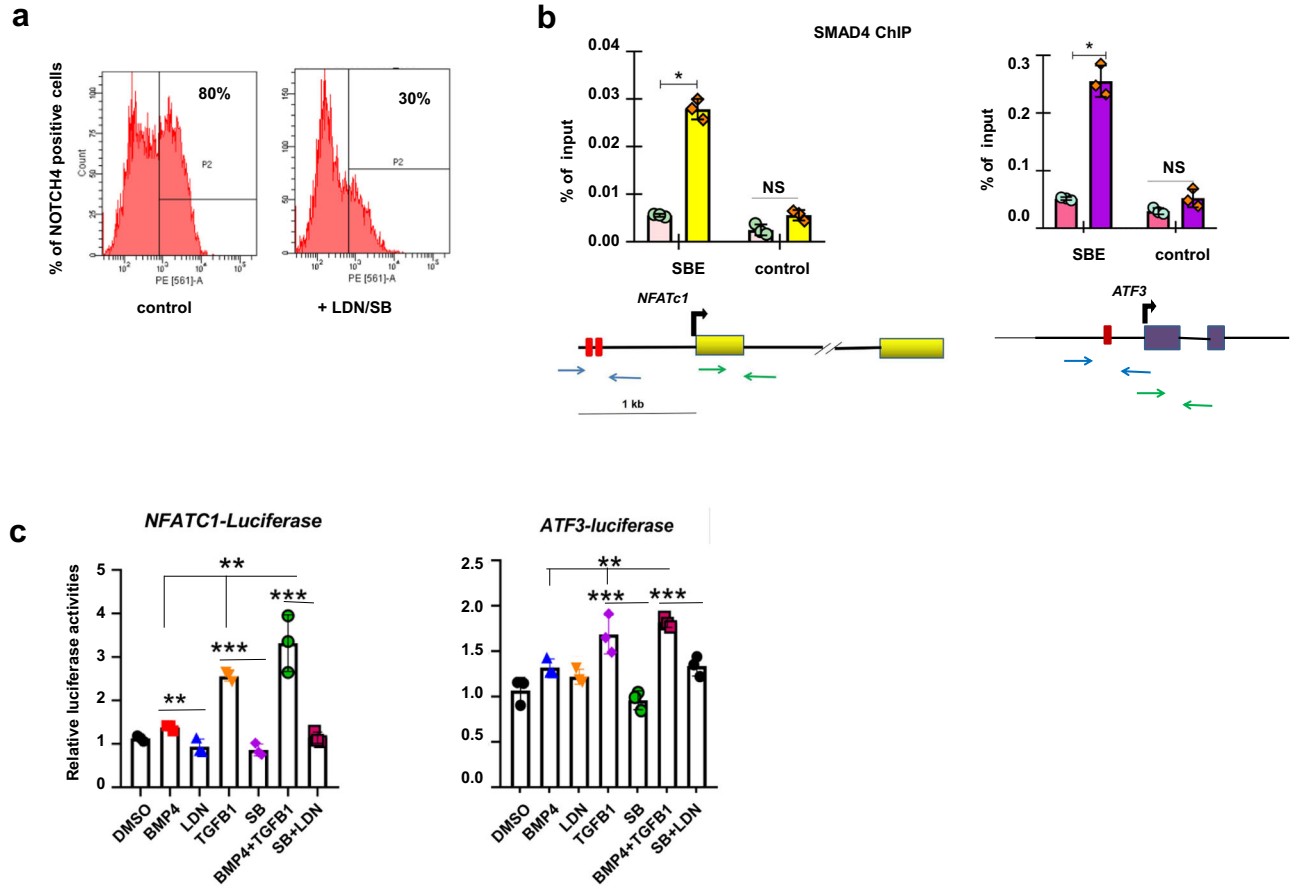

**Fig. 5 BMP4 and TGFb1 induce VEC fate by promoting *NFATc1/ATF3*. a** Flow cytometry analysis of day 7 hPSC-derived VELs in the presence and absence of SB and LDN, showing the percentage of NOTCH4 positive cells. **b** ChIP experiments showing the enrichment of SMAD4 at *NFATc1* and *ATF3* promoters. **c** Luciferase reporter assay showing *NFATc1* and *ATF3* promoter activities in the presence and absence of LDN and SB. Experiments were repeated at least two times. The paired *t* test in Graphpad software was used for the statistical analysis. Significant levels are: *$p < 0.05$; **$P < 0.01$; ***$P < 0.001$.

well as less-characterized genes such as *ATF3*, *BMP4*, *GJA5*, *SELE*, *KLFs*, *THBD*, *PALMD*, *ITM2A*, *FGD5*, *CXCL12*, *LDB2*, *CDH11*, *CDH13*, *GALC*, *ISLR*, *HMCN1*, and *TSPAN8*. Of which, *CDH11* and *CXCL12* have been shown to be implicated in ECC/VEC formation[46,47]. The qRT-PCR analysis of selected genes confirmed the RNA-seq results (Fig. 4c; Supplementary Fig. 4d, e). WB results showed that BMP4 was expressed at higher levels in hPSC-derived VELs and the primary VECs than HUVEC (Fig. 4d). IF results showed that LDB2, GALC, and ITM2A were more highly expressed in hPSC-derived VELs than HUVEC (Fig. 4e). Similar results were observed for CXCL12 and FGD5 (Supplementary Fig. 4g). ITM2A, a single-pass type II membrane protein, was of particular interest to us. Flow cytometry results showed that approximately 77% of day 7 hPSC-derived VELs were ITM2A/VE-cad double positive (Supplementary Fig. 4h).

To further investigate how hPSC-derived VELs resemble the primary VECs, we re-analyzed our scRNA data from two healthy aortic valves from around 40-year old[48]. Based on the expression of side-specific VEC genes such as *VCAM1* and *MGST1*[49], two VEC subclusters were identified which may represent the aortic and ventricular side of VECs (Fig. 4f). Again, we found that VECs from 40-year-old abundantly expressed markers *ENG*, *SELP*, *NFkB1*, *DKKs*, and *KLFs* (Fig. 4g, h). The qRT-PCR analysis further confirmed that they were abundantly expressed in both aortic and ventricular side of VECs (Fig. 4i).

Finally, we examined the expression of oscillatory shear-related genes such as *KLF2*, *CAV1*, *NOS3*, *EDN1*, *BMP4*, *CTSK*, and *THBS1*[32]. RNA-seq results showed that these genes were expressed

at lower levels in day 7 hPSC-derived VELs than in primary VECs. We confirmed this by qRT-PCR analysis (Supplementary Fig. 4i). Thus, although hPSC-derived VELs resembled the primary VECs, they might be still in an immature state, presumably due to lack of hemodynamic stimulus under normal physiological conditions (see discussion).

**The combined treatment with BMP4 and TGFb1 promotes VEC fate by activating NOTCH signaling and enhancing NFATc1/ATF3.** So far, we have shown that the crosstalk between BMP and TGF-β signaling pathways is important for directing hPSC-derived CPCs to the VEC-like fate. Remarkably, core NOTCH-related genes such as *NOTCH1/4*, *DLL3/4*, *JAG1/2*, and *HEY1/2*, as well as NOTCH target genes such as *FOS*, *NR4A2*, *CD44*, *CDKN1A*, *NFKB1*, *CCDC141*, *GATA2*, and *HAND2*, were upregulated during VEC induction (Fig. 2c; Fig. 3d; Supplementary Fig. 3f). As NOTCH genes such as *NOTCH1/4*, *JAG1*, and *HEY1/2* have been shown to be targets of TGF-β/BMP signaling[12,50–52], we asked whether NOTCH signaling was activated by the combined treatment with BMP4 and TGFb1. When BMP inhibitor LDN and/or TGF-β inhibitor SB were applied, the expression of NOTCH signaling components and its target genes was greatly reduced (Supplementary Fig. 5a, b). Flow cytometry results showed that the percentage of NOTCH4 positive cells was greatly decreased in day 7 cell cultures when CPCs were treated with LDN/SB (30% in treated vs 80% in control) (Fig. 5a). Thus, the combined treatment with

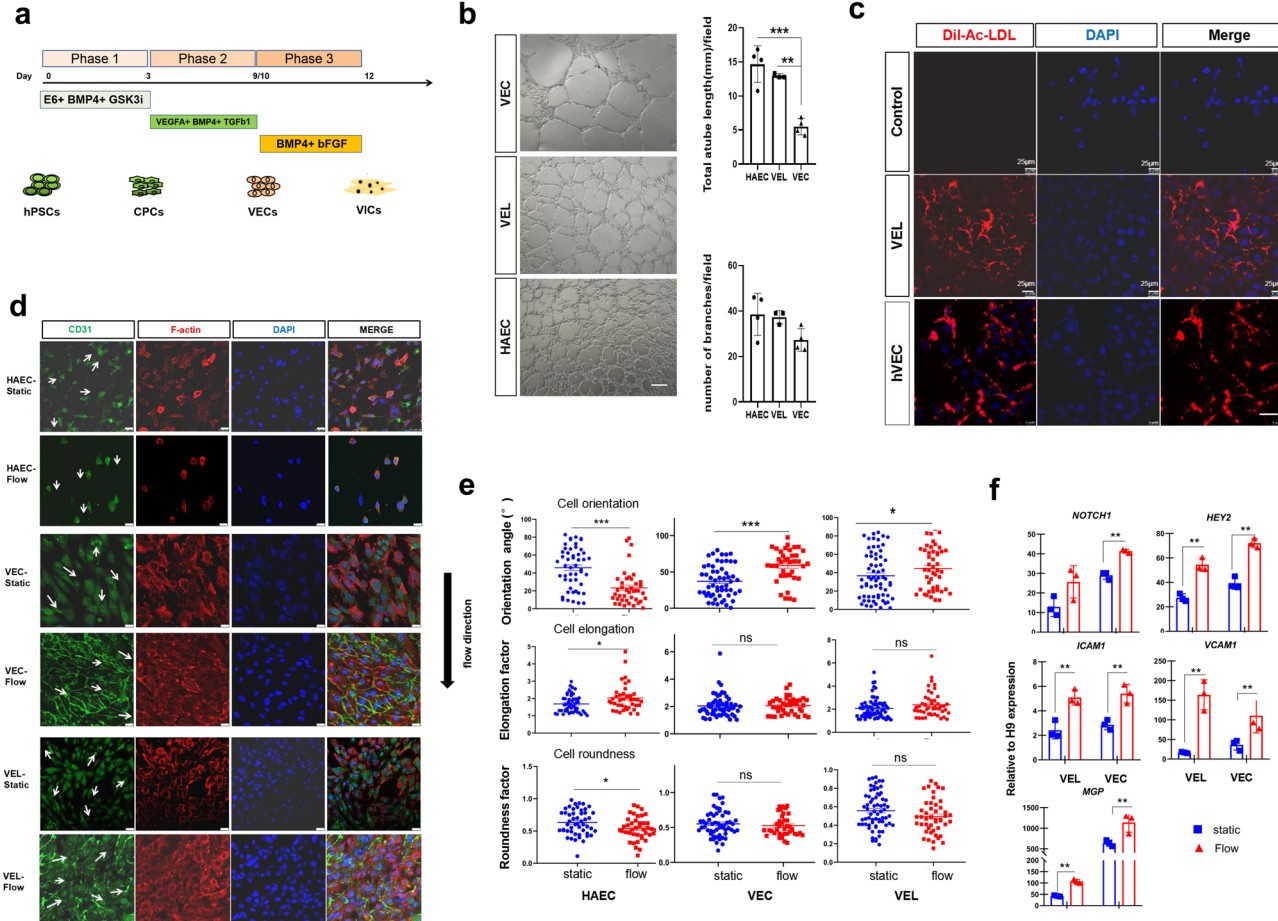

**Fig. 6 Functional characterization of hPSC-derived VELs. a** Cartoon depicting the protocol that is used to differentiate hPSC to VELs in 10 days. **b** Tube formation assay of day 7 hPSC-derived VELs, the primary VECs and HAEC. Left: representative images; right: quantification of capillary-like structures in terms of branches and tube length by ImageJ. Scale bar: 25 μm. **c** Fluorescence signals showing that hPSC-derived VELs could uptake Ac-LDL, similar to primary VECs. **d** Confocal images showing the alignment of hPSC-derived VELs, the primary VECs from 16-year old and HAEC after exposed to fluid flow for 48 hpf. Fluid flow direction is shown by the black arrow. White arrows show the direction of individual cells. Scale bar: 25 μm. **e** Quantification of cell shape and orientation for primary VECs, VELs, and HAEC, under static and flow conditions (**d**). Cell shape and orientation were quantified by ImageJ. At least 50 cells were evaluated per condition across three independent replicates. Graph shows mean as horizontal bar. **f** The qRT-PCR analysis of indicated genes for VECs and VELs under static and flow conditions. All experiments were repeated three times. The paired *t* test in Graphpad software was used for the statistical analysis. Significant levels are: \**p* < 0.05; \*\**P* < 0.01; \*\*\**P* < 0.001. Shown are representative images in (**b–e**).

BMP4 and TGFb1 may induce the VEC fate at least partially by activating NOTCH signaling.

We found that a panel of transcription factors such as *ATF3* and *KLFs* were upregulated during hPSC differentiation into VELs. *Atf3* has recently been shown to be highly expressed in the OFT where aortic valve is originated[53]. We speculated that combined treatment with BMP4 and TGFb1 may promote VEC fate by inducing *NFATc1* and *ATF3*[54]. SMAD transcription factors, including the R-SMADs and common SMAD (SMAD4), are the mediators of signal transduction by TGF-β superfamily which includes BMP and TGF-β[55,56]. We identified two SBEs composed of sequence CAGACA and one SBE in the proximal promoters of human *NFATc1* and *ATF3* genes (Fig. 5b). The ChIP-PCR analysis showed that SMAD4 binding was enriched at promoter regions of *NFATc1* and *ATF3* (Fig. 5b). Next, the luciferase reporter plasmids containing −2 kb *NFATc1* and *ATF3* promoters were constructed and subsequently transfected into HEK293T cells in the presence and absence of BMP4 and TGFb1 or the inhibitors of the respective signaling pathways. The luciferase reporter assay showed that BMP4 and TGFβ1 strongly induced while LDN and SB repressed the luciferase activities (Fig. 5c).

Taken together, we propose that combined treatment with BMP4 and TGFb1 may induce the VEC fate at least partially by activating NOTCH signaling and by enhancing *NFATc1/ATF3*.

**Functional characterization of hPSC-derived VELs.** Our final protocol for hPSC differentiation into VELs is illustrated in Fig. 6a. HPSC-derived VELs could be passaged up to 4 times in vitro with commercial EBM2 medium before eventually adopting a fibroblastic morphology (Supplementary Fig. 6a). At passage 3, hPSC-derived VELs abundantly expressed EC marker CD31 and barely expressed α-SMA. At passage 4, a subset of cells expressed both CD31 and α-SMA, indicative of undergoing differentiation. At passage 5, the majority of cells downregulated the expression of CD31, suggesting loss of EC phenotype (Supplementary Fig. 6b).

We next investigated whether hPSC-derived VELs functionally resembled the primary VECs. HPSC-derived VELs formed vascular network-like structures on Matrigel, similar to primary VECs and HAEC (Fig. 6b), and abundantly expressed the endothelial markers and our newly-identified candidate markers such as CXCL12 and LDB2 (Supplementary Fig. 6c, d). A

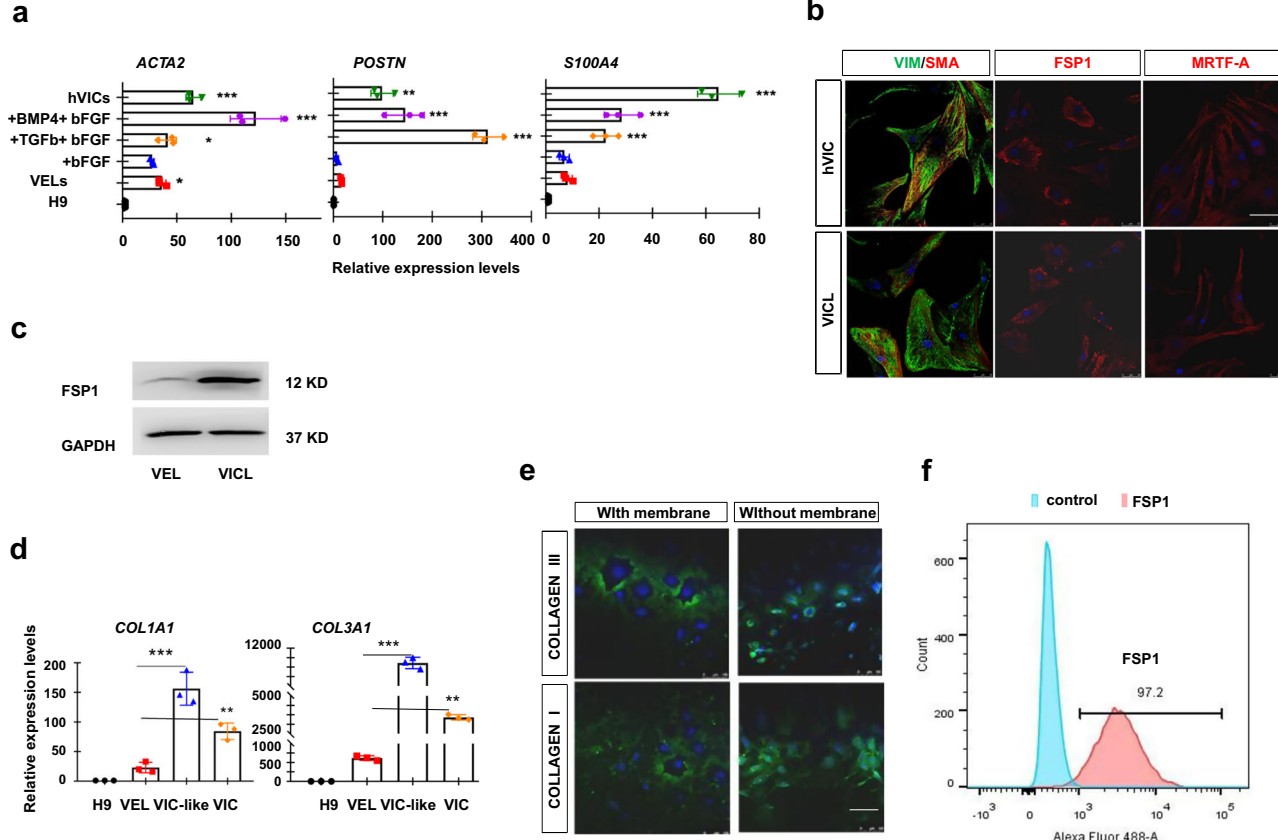

**Fig. 7 HPSC-derived VELs to VIC-like (VICL) cells by inducing EndoMT. a** The qPCR analysis of indicated VIC markers for hPSC-derived VELs that were treated with the indicated signaling molecules for 3–6 days. **b** IF staining of hPSC-derived VIC-like (VICL) cells showing the expression of the indicated VIC markers. The primary VICs were used as positive controls. Scale bar: 100 μm. **c** WB analysis showing the expression levels of FSP1 in hPSC-derived VEC-like cells and VICL cells. **d** The qRT-PCR analysis of type I collagen marker *COL1A1* and type III collagen marker *COL3A1* in hPSC-derived VIC-like cells. The primary VICs were used as positive controls. **e** IF staining showing the expression levels of COLLAGEN I and III in hPSC-derived VIC-like cells, with and without membrane breaking treatment. **f** Flow cytometry analysis showing the percentage of FSP1 positive cells in hPSC-derived VIC-like cells. All experiments were repeated three times. The paired *t* test in Graphpad software was used for the statistical analysis. Significant levels are: *$p < 0.05$; **$P < 0.01$; ***$P < 0.001$. Shown are representative images in (**b, c, e**).

hallmark of ECs is the uptake of LDL[32]. To assess the LDL uptake, hPSC-derived VELs and the primary VECs were incubated with Alexa Fluor 594 conjugated to acetylated LDL (Ac-LDL) for 2 h. The confocal images showed that both cell types were able to robustly incorporate Dil-Ac-LDL as indicated by the red fluorescence in each cell (Fig. 6c).

To investigate how hPSC-derived VELs response to the fluid flow, cells were plated on the μ-Slide (ibidi, Germany) and subjected to 15 dynes/cm² flow shear stress for up to 48 h. The cells were then fixed and subjected to double IF staining with CD31 and F-actin antibodies (Fig. 6d). The primary VECs isolated from 15-year old and HAEC were used as controls. In static condition, all three cell types are randomly oriented, with the angle of orientation ranging from 0° to 90°. Under flow condition, the average angle of deviation from the flow direction for HAEC was around 15°, showing a nearly parallel orientation. The primary VECs, however, exhibited a perpendicular orientation, with an average angle of around 70°. The orientation of hPSC-derived VELs partially resembled that of primary VECs (Fig. 6e; Supplementary Fig. 6e). Next, we compared the cell shape for three cell types by measuring cell elongation and roundness factors. In static condition, HAEC, the primary VECs and hPSC-derived VELs have an average cell roundness factor of 0.65, 0.58, and 0.56, respectively. After 48 h of flow, these values were reduced to 0.52, 0.53, and 0.52, respectively. Consistently, in

static condition, HAEC, the primary VECs and hPSC-derived VELs have an average cell elongation factor of 1.8, 2.0, and 2.0, respectively. After 48 h of flow, these values increased to ~2.3. Thus, alteration of cell shape index showed that all three cell types became more elongated after 48 h of flow, with hPSC-derived VELs more closely resembling the primary VECs (Fig. 6e).

Calcific aortic valve sclerosis involves inflammatory processes and occurs on endothelialized valve leaflets[32]. We investigated the acute effects of flow shear stress on expression of *VCAM-1* and *ICAM-1*, two inflammatory markers, by exposing hPSC-derived VELs and the primary VECs to shear stress for 2 days. The qRT-PCR analysis showed that *VCAM1* and *ICAM1* were induced by the shear stress in both HPSC-derived VELs and the primary VECs (Fig. 6f). It has been shown that shear stress activates *NOTCH1* and its target genes *HEY2* and *MGP* in primary human aortic VECs[57]. Similar to primary VECs, hPSC-derived VELs increased the expression of *NOTCH1* and its target genes *HEY2* and *MGP* in response to the shear stress (Fig. 6f).

**Conversion of hPSC-derived VELs to VIC-like cells by inducing EndoMT.** During valvulogenesis, valvular interstitial cells (VICs) are mainly derived from endocardial cushion cells through an EndoMT process[1,10,58]. We investigated whether hPSC-derived VELs could be converted to VIC-like (VICL)

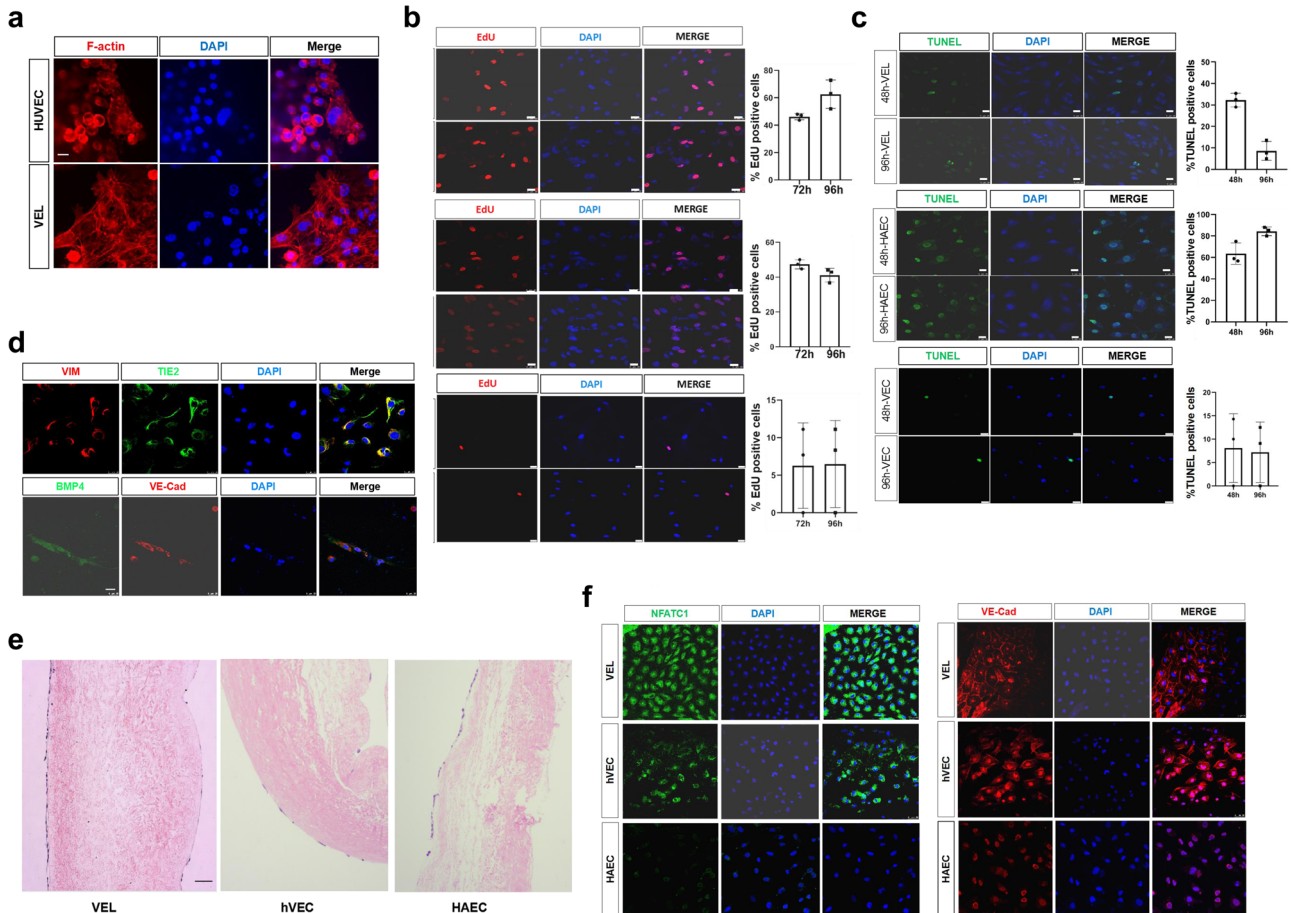

**Fig. 8 Endothelialization of de-cellularized porcine heart valves by hPSC-derived VELs. a** IF staining of F-actin showing the morphology of hPSC-derived VECs and HUVEC seeded onto the de-cellularized porcine heart valves. Scale bar: 25 μm. **b** Left panel: EdU staining results for hPSC-derived VELs, the primary VECs from 15-year old and HAEC that have been seeded onto the de-cellularized porcine heart valves for 72 and 96 h. Right panel: bar graphs showing the percentage of EdU positive cells for left panel. Bar: 25 μm. **c** Left panel: TUNEL assay for hPSC-derived VELs, the primary VECs and HAEC that have been seeded onto the de-cellularized porcine heart valves for 48 and 96 h. Right panel: bar graphs showing the percentage of TUNEL positive cells for left panel. Bar: 25 μm. **d** IF staining images showing BMP4/VE-Cad and VIM/TIE2 double positive signals in hPSC-derived VELs seeded onto the surface of the de-cellularized porcine heart valves. Bar: 50 μm. **e** HE staining showing the lining of hPSC-derived VELs at the surface of DCVs after 2–3 weeks of co-culturing. Lateral view of the representative DCVs. Left: the aortic side of the valve; Right: the ventricular side of the valve. Bar: 100 μm. **f** IF staining images showing NFATc1 and VE-cad signals in hPSC-derived VELs, the primary VECs and HAEC that have been seeded on the surface of de-cellularized porcine heart valves for 2–3 weeks. Scale bar: 100 μm. All experiments were repeated three times. Shown are representative images in (**a**–**f**).

cells through EndoMT by treating them with higher concentration of TGFb1 and bFGF over a time course of 6 days (Fig. 7a; Supplementary Fig. 7a). HPSC-derived VELs quickly lose the cobble-stone morphology, and displayed the fibroblastic morphology (Fig. 7b). After 3 days treatment, VIC-type markers such as *POSTN, ACTA2,* and *S100A4/FSP1* were strongly induced (Fig. 7a; Supplementary Fig. 7b), accompanied by upregulation of EndoMT genes such as *SNAI1, TWIST1, SOX9, AGGRECAN, TENASCIN,* and *SCLERAXIS*[58] (Supplementary Fig. 7c). We further confirmed these by IF staining and WB analyses (Fig. 7b, c).

Next, we investigated whether the VIC-like cells expressed extracellular matrix components (ECM). We found that ECM-related genes such as *COLLAGEN I* (*COL1A1*) and *III* (*COL3A1*) were greatly induced when hPSC-derived VELs were treated with bFGF and TGFb for 6 days (Fig. 7d). The IF results showed that day 6 treated cells abundantly expressed COLLAGEN I and III (Fig. 7e). Flow cytometry analysis revealed that the vast majority of the cells (>97%) were FSP1 positive (Fig. 7f). Based on the above data, hPSC-derived VELs can be converted to valvular interstitial-like cells by inducing EndoMT.

**HPSC-derived VELs interacts with de-cellularized porcine heart valves**. We have previously made poly(ethylene glycol) tetraacrylate (PEG-TA) cross-linked de-cellularized porcine aortic valves (PEG-DCVs)[2]. Although the acellular PEG-DCVs exhibit improved mechanical and anti-calcification properties compared to the glutaraldehyde cross-linked counterparts, the poor clinical long-term result remains unsatisfactory likely due to the early tissue degeneration[3].

We believe that hPSC-derived VELs may serve as a good source of seed cells for the next-generation TEVs. To initially evaluate this possibility, we studied the growth and adherent properties of hPSC-derived VELs seeded on the modified DCVs. The primary VECs from 15-year old, HAEC and HUVEC were included for comparison. We found that 1 day after seeding, about 50% of VELs attached to the surface, similar to HUVEC as revealed by IF staining of F-actin, CD31, and VE-cad (Fig. 8a; Supplementary Fig. 8a, b). Cell proliferation assay by EdU staining showed that hPSC-derived VELs were highly proliferative, with 43% of cells being EdU positive at day 3 and 60% of cells being EdU positive at day 4 (Fig. 8b). By contrast, the primary VECs showed much slower cell

proliferation, with only 5% of cells being EdU positive at day 3 and 4.

Next, we accessed the cell apoptosis by TUNEL staining. We found that hPSC-derived VELs and VECs exhibited less cell apoptosis compared to HAEC. For instance, ~30% of hPSC-derived VELs were TUNEL positive at day 2, but quickly reduced to around 7% at day 4 (Fig. 8c). By contrast, more than 60% of HAEC were TUNEL positive at day 2 and day 4. Thus, hPSC-derived VELs exhibited higher cell proliferation rate and less cell apoptosis. IF staining results showed that the seeded hPSC-derived VELs at day 4 co-expressed TIE2 and VIM as well as VE-Cad and BMP4 (Fig. 8d). BMP4 has been shown to be abundantly expressed in primary VECs in both physiological and shear stress conditions[32].

To understand the long term interaction between hPSC-derived VELs and the DCVs, we seeded the VELs at a low density and cultured them for 2–3 weeks. We found that hPSC-derived VELs were well attached, displaying a monolayer of endothelium on either side of the DCVs, as revealed by HE (hematoxylin-eosin) staining of cross-sectioned DCVs (Fig. 8e). By contrast, HAEC and the primary VECs were unevenly distributed, with much smaller number of cells. This was also confirmed by IF staining of the surface of DCVs using NFATc1 and VE-Cad antibodies (Fig. 8f).

Together, the results demonstrated that endothelialization occurred and cell proliferated after co-culture of DCVs and hPSC-derived VELs, and that hPSC-derived VELs exhibited superior proliferative and clonogenic potential than the primary VECs and HAEC.

## Discussion

The role of TGFβ, BMP, and NOTCH signaling in cardiac cushion and valve formation has been well discussed using different models[12,15,59]. Despite the great progress made in the field, there are a number of outstanding questions. For instance, what are the phenotypical and genetic differences between the endocardial cushion cells (ECCs) compared to the neighboring endocardial cells? How the ECCs are defined and specified within the endocardium? What controls valve endocardial cell fate determination and which signaling pathways are involved in the process and what are the key downstream targets or mediators? What is the phenotypical and genetic differences between the ECCs and VECs? How the VEC identity is maintained throughout lifetime? Answering these key questions will undoubtedly deepen our understanding of valvulogenesis and valve biology, and benefit the treatment or prevention for valve diseases. As our in vitro hPSC VEC differentiation recapitulates embryonic valvulogenesis, it may serve as a unique system to tackle some of the above-mentioned questions. First, we found that the crosstalk or synergy of BMP and TGF-β signaling pathways is important and sufficient for the specification of VEC fate via an intermediate ECC stage[58]. Second, we precisely dissected the requirement of key signaling pathways and transcription programs for the determination of VEC cell fate. We found that NOTCH signaling is quickly induced at early stage, and WNT/BMP signaling pathways and a panel of TFs such as NFATc1, HEY2, and HAND2 are activated at late stage as a part of VEC formation (Figs. 2c, 3e). As NOTCH signaling is activated downstream of TGFβ signaling, it may interpret why addition of DLL4 does not augment the expression of ECC genes, in the presence of VAGFA/BMP4/TGFb1. We propose that the VEC identity is established and maintained by an integrated network consisting of VEC-specific transcription factors (such as NFATC1/HAND2/HEY2/ATF3/KLFs) and key signaling pathways (such as WNT/BMP). Third, by performing transcriptome assay of hPSC-derived VELs with VECs from embryos of 5–25W and adults, the genetic

signatures of ECCs and VECs were better defined, at single cell and bulk levels. Fourth, our hPSC VEC differentiation largely recapitulates the key events of in vivo valvulogenesis. In the future, this unique system will allow more in depth mechanism study of valvulogenesis.

Our vitro hPSC-derived VELs may still represent an immature form of VECs, as revealed by expression levels of shear stress response genes such as KLF2, CAV1, NOS3, and ICAM1 as well as functional assays. This is likely because the described protocol lacks of mechanical stimulus that are exposed to hVECs in normal physiological conditions. Differentiating hPSCs into VELs under conditions that mimic physiological stimulation is highly desirable. It has been suggested that EC identity could be better maintained under continuous stimulation with shear stress or cyclic strain by using bio-reactors in vitro or by grafting the vessel into a host organism in vivo[5]. Also, there is strong evidence that neural crest cells (cNCCs) contribute to the cardiac valves[41,60]. Our current protocol is entirely based on mesodermal lineage, which may neglect the contribution of cardiac NCCs.

Our long term goal is to make autologous stem cell-based valve organoids with self-renewal and self-assembly capacities after transplantation that could substitute the current mechanical and bioprosthetic valves. The ability to generate both VEC-like and VIC-like cells from hPSCs and the proof-of-concept study of this work predict a promising future of making autologous stem cell-based valve organoids.

## Methods

**Isolation of human VECs**. Healthy aortic valve tissue specimens were harvested from patients undergoing repair of aortic dissection requiring aortic valve replacement. Side-specific human aortic VECs [from the aortic side (VEC-A) and the ventricular side (VEC-V)] were isolated (according to an Institutional Review Board approved protocol at UNION hospital, HUST) using a brief collagenase digestion and gentle scraping method[32]. Briefly, Human aortic VECs from 7-, 9-, 15-, 20-, 30-, 40-, 50-year old were isolated and cultured with ECGM or EBM-2 medium (Lonza) for 1–2 days to examine the cell morphology. To keep the intact features of the isolated VECs, the cells were often not passaged and used directly for bulk RNA-sequencing or scRNA.

The quality of isolated hVECs was evaluated by three criteria. First, the primary VECs were checked morphologically under the microscope. The batch of the isolated VECs with contamination with fibroblast-like cells (likely the VICs) was not used for RNA-sequencing. The vast majority of the primary VECs (>99%) adopted a cobblestone morphology. Second, to reduce the possibility that the isolated VECs were contaminated by the VICs or other non-valvular cell types, cells with fibroblastic morphology were manually removed. Third, the RNA-seq data were used to double-check the quality of the isolated primary VECs. The RNA-seq results showed that definitive EC markers CDH5 and PECAM1 were highly expressed while the fibroblast markers S100A4 and POSTN, as well as cardiomyocyte markers TNNT1/2, were barely detectable (RPKM < 5; P < 0.05).

Human endothelial cell line HUVEC (#CRL-1730) was purchased from the American Type Culture Collection (Rockville, MD, USA). Human aortic endothelial cell line HAEC (#6100) was purchased from Shanghai Cell Bank, CAS (Shanghai, China).

**Maintenance of human PSCs**. Human PSCs including human induced pluripotent stem cells[26] and PGP1 human induced pluripotent stem cells (kindly provided by Prof. Zhang from Hubei University, Wuhan, China), and human ESC lines (H8 and H9 lines, Wicell, WI, USA) were used for this study. HPSCs were maintained with mTsSR1 medium (STEMCELL Technologies, Canada) or E8 medium (Life Technologies, USA), in feeder-free plates coated with Matrigel (Corning)[25]. Briefly, HPSCs were treated with Accutase (Gibco) for 4 min, and single hPSCs were centrifuged at $400 \times g$ for 4 min, and seeded onto the Matrigel-coated 6-well plates at 30,000 cell/cm$^2$ in mTeSR1 supplemented with 0.1 µM ROCK inhibitor Y-27632 (Tocris, USA). HPSCs were passaged every four days[61,62].

**CPC differentiation**. We have described the method for the generation of CPCs from hPSCs[25]. To optimize the generation of ISL1$^+$ KDR$^+$ CPCs, we further modified the method. HPSCs were passaged as described above and reseeded at $5 \times 10^5$ cell/cm$^2$ (H9) or $8 \times 10^4$ cell/cm$^2$ (H8 and iPSCs) onto the Matrigel coated plates with Essential 6 medium (Gibco) supplemented with 25 ng/ml WNT3a (5036-WN, R&D Systems) or CHIR99021 (0.6 µM) and 100 ng/ml BMP4

**11**

**Table 2 The primers used for qRT-PCR analysis.**

| Genes | Forward (5′–3′) | Reverse (5′–3′) |
|---|---|---|
| GAPDH | TGTTGCCATCAATGACCCCTT | CTCCACGACGTACTCAGCG |
| NANOG | TCCTGAACCTCAGCTACAAACA | GGTAGGTGCTGAGGCCTTCT |
| POU5F1 | GTGGGGGCAGGGGAGTTTGG | AGTGTGTCTATCTACTGTGTCCCAGGC |
| T | GGGTGGCTTCTTCCTGGAAC | TTGGAGAATTGTTCCGATGAG |
| CXCR4 | CGCCTGTTGGCTGCCTTA | ACCCTTGCTTGATGATTTCCA |
| PDGFα | GATTAAGCCGGTCCCAACCT | GGATCTGGCCGTGGGTTT |
| PDGFβ | TGGCAGAAGAAGCCACGTT | GGCCGTCAGAGCTCACAGA |
| GATA4 | TCCAAACCAGAAAACGGAAGC | GCCCGTAGTGAGATGACAGG |
| VEGFR2 | CGGCTCTTTCGCTTACTGTT | TCCTGTATGGAGGAGGAGGA |
| SOX1 | GCGGTAACAACTACAAAAAACTTGTAA | GCGGAGCTCGTCGCATT |
| NKX2.5 | CCAAGGACCCTAGAGCCGAA | ATAGGCGGGGTAGGCGTTAT |
| MEF2C | ATGGATGAACGTAACAGACAGGT | CGGCTCGTTGTACTCCGTG |
| ISL1 | GCAAATGGCAGCGGAGCCCA | AGCAGGTCCGCAAGGTGTGC |
| FOXF1 | GTACCCGCACCACGACAGCTC | ATACCGCGGGATGCCTTGCAG |
| PECAM1 | GCAACACAGTCCAGATAGTCGT | GACCTCAAACTGGGCATCAT |
| ETV2 | AACACCAGCTGGGACTGTTC | GAGGTTTGACCGGGAATTTT |
| NFATc1 | GCATCACAGGGAAGACCGTGTC | GAAGTTCAATGTCGGAGTTTCTGAG |
| eNOS | CCAGCTAGCCAAAGTCACCAT | GTCTCGGAGCCATACAGGATT |
| POSTN | CTGCCAAACAAGTTATTGAGCTGGC | AATAATGTCCAGTCTCCAGGTTG |
| a-SMA/ACTA2 | TTTCCGCTGCCCAGAGAC | GTCAATATCACACTTCATGATGCTGT |
| CDH5 | TGTTCACGCATCGGTTGTTC | ACTTGGTCATCCGGTTCTGG |
| S100A4/FSP1 | GATGAGCAACTTGGACAGCAA | CTGGGCTGCTTATCTGGGAAG |
| VIMENTIN | GAAGGCGAGGAGAGCAGGATT | CAAGGTCATCGTGATGCTGAG |
| HAND2 | TACCAGCTACATCGCCTACCT | TCACTGCTTGAGCTCCAGGG |
| CXCL12 | GGGCTCCTGGGTTTTGTATT | GTCCTGAGAGTCCTTTTGCG |
| GATA5 | CCTGCGGCCTCTACCACAA | GGCGCGGCGGGACGAGGAC |
| CD34 | CAACCAGGGGAGCTCAAGTT | AAGACACTACTCGGCTTGGC |
| JAG1 | TCAGTCGGGAGGCAAAT | GCCACCGTTTCTACAAGG |
| MCAM | TCTCCCAGTCCCAAGGC | CTCCCCAGGTTCGCTCT |
| JAG2 | GACACCAATCCCAACGACT | TAGGCATCGCACTGGAAC |
| GJA4 | CGACCAGTACGGCAACAA | TCATCGCAGAACCTCCCT |
| HEY2 | TGGGGAGCGAGAACAAT | TCAAAAGCAGTTGGCACA |
| TNNT1 | GCTGGAAGTGAGGATGCC | GCTACCGATGGGACAAACA |
| IGFBP3 | AGCGGGAGACAGAATATGG | TTTGGAAGGGCGACACT |
| FGF13 | TCTGCGAGTGGTGGCTA | CCTGACTGCTGCTGACG |
| PITX1 | TGTCGTCGCAGTCCATGTT | GGAGCCGGTGAGGTTGTT |
| vWF | CCTTGAATCCCAGTGACCCTGA | GGTTCCGAGATGTCCTCCACAT |
| SMAD6 | CACTGAAACGGAGGCTACCAAC | CCTGGTCGTACACCGCATAGAG |
| COLIA1 | GATTCCCTGGACCTAAAGGTGC | AGCCTCTCCATCTTTGCCAGCA |
| TGFb | TACCTGAACCCGTGTTGCTCTC | GTTGCTGAGGTATCGCCAGGAA |
| TGFb2 | AAGAAGCGTGCTTTGGATGCGG | ATGCTCCAGCACAGAAGTTGGC |
| CTSK | GAGGCTTCTCTTGGTGTCCATAC | TTACTGCGGGAATGAGACAGGG |
| EMCN | ACAATTCCAGAAAACACCTCA | CACATTCGGTACAAACCCA |
| MESP1 | CTCTGTTGGAGACCTGGATG | CTCAAACCTGCTTGCGTG |
| EDN1 | CTACCTCACCTATATTGCACT | GACCAGACTTCTACGAGGCTA |
| NES | TCCTACAGCCTCCATTCTT | GCAGCACTCTTAACTTACG |
| NOTCH4 | AGTGGCAGAAATAGGAGGG | ATTCCCACTGCCTCCAGAC |
| DDR1 | AGCTCCTGGTCAGATTCCAC | GATCCACCTGCAAGTACTCCT |
| NFATC4 | TCCCTTCAGCATCGGCAAC | AAGCCTTCTGATAGGTAAGGAGT |
| TAL1 | CGATCCCAGTTGGAGGGTTC | CCAGTCCAGGGAATCGCAAA |
| GAL | GTGCTGTAACCTGAAGTCA | ACAGGAATGGCTGACTCT |
| DLL4 | GGACCAGGAGGATGGCT | GCTCAAGGTGCTGTGTTCA |
| LEFTY1 | GCCCTGAATTTGCTTCCTC | GACACATTGGGCTTTCTGC |
| BMP4 | CTGGTCTTGAGTATCCTGAGCG | TCACCTCGTTCTCAGGGATGCT |
| LDB2 | TTCCACCAGCAGCACTTCCAAC | TCAGAGTTGGCTCTCCTACCAC |
| FGD5 | CCTTGTCATCGCACAGGAACTG | CTCTGCCTTCATGGTCCATGTC |
| BGN | TTGAACCTGGAGCCTTCGATGG | TTGGAGTAGCGAAGCAGGTCCT |

(314-BP, R&D Systems) for 1 day. Next, cells were cultured in E6 medium supplemented with 50 ng/ml BMP4 and 20 ng/ml bFGF for 3 days with daily medium change.

**CPCs to VEC-like cells**. HPSC-derived day 3 CPCs were dissociated and seeded at $1 \times 10^5$ cells/cm² onto cell culture dishes. Cells were cultured for additional 6–9 days in E6 medium supplemented with 100 ng/ml VEGF, 10 ng/ml BMP4 and 10 ng/ml TGFb1. The medium was changed every day.

**Bulk RNA-sequencing and heat map analysis**. HAEC, HUVEC, H9 (WA09 hESCs) cells, HFF (human foreskin fibroblasts), the primary VICs, the primary VECs, and hPSC-derived VELs were collected and lysed with 1 ml Trizol (Transgen Biotech, China). RNA sample quality was checked by the OD 260/280 value using the Nanodrop 2000 instrument. When necessary, hPSC-derived VELs were sorted with VE-cad or CD31 magnetic beads (Miltenyi Biotec, GmbH, Germany) before RNA-sequencing. RNA samples were sent to the BGI China where RNA-sequencing libraries were constructed and sequenced by a BGI-500 system. RNA-seq experiments were repeated at least 2 times. Differentially expressed genes

(DEGs) were defined by FDR < 0.05 and a Log2 fold change> 1 was deemed to be DEGs. The heat map was constructed based on the commonly expressed genes in different cell types, and the top 100 DEGs were listed as part of the heap map. Gene ontology (GO) analysis for differentially expressed genes (DEGs) and heat maps were generated from averaged replicates using the command line version of deepTools2.

**Single cell RNA-sequencing analysis**. Two healthy aortic valve specimens of 40-year old were processed according to the manufacturer's instructions (Document CG00055, 10X genomics) and single-cell gene expression profiles determined by Chromium Single Cell 3′ (v2 Chemistry). Single Cell 3′ v2 libraries were generated using Single Cell 3′ v2 Reagent Kits according to the manual. The libraries were sequenced on the Illumina HiSeq Xten platform. Data were processed using Cell Ranger (v2.0.1) software. Cell counts were then used to map the reads to a reference genome (hg19) using STAR Aligner (v2.5.1b43). A digital gene expression matrix was 7 constructed from the raw sequencing data as described above. Downstream analyses were performed using Monocle 2 v.2.6.023 software.

The original scRNA data for embryonic cardiac cells were downloaded and re-analyzed[36]. When we classified all filtered cells, we utilized the Seurat R package to obtain highly variable genes, which were used to perform principle component analysis (PCA). After that, significant PCs were identified with the JackStraw function in the Seurat R package, and were selected as the significant components for t-SNE analysis. We set the clustering parameter resolution to 0.3 and perplexity to 100 for the function FindClusters in Seurat. When we identified subpopulations of each cell type, we performed second-level clustering with the clustering and classification algorithm with default parameters. We used Monocle2 and UMI to perform the pseudotiming analysis, firstly by obtaining variable genes, followed by calculating the pseudotime of cells.

For embryonic heart cells, we analyzed 2347 genes (average expression >10, and FDR < 0.01) that are dynamically expressed along the pseudotime trajectory[36]. For the heatmap analysis, we chose the top 2000 genes (average expression >10, and FDR < 0.01), and divided them into two groups.

For 40-year old VEC cells, we extracted the VEC clusters (total 746 cells) from the original scRNA data[48], and re-analyzed them. Briefly, using Seurat R package, we performed clustering analysis and obtained 2000 variable genes by which we used to perform PCA analysis. By JakStraw in the Seurat package, we identified important PCs, and selected 15 PCs for UMAP analysis. Next, using FindClusters, we determined the two VEC subclusters.

**Quantitative real-time PCR**. Total RNA for cells was extracted with a Total RNA isolation kit (Omega, USA). 1 μg RNA was reverse transcribed into cDNA with TransScript All-in-One First-Strand cDNA synthesis Supermix (Transgen Biotech, China). Quantitative real-time PCR (qRT-PCR) was performed on a Bio-Rad qPCR instrument using Hieff qPCR SYBR Green Master Mix (Yeasen, China). The primers used for RT-qPCR are listed in Table 2, Supplementary Table 2. All experiments were repeated for three times. The relative gene expression levels were calculated based on the $2^{-\Delta\Delta Ct}$ method. Data are shown as means ± S.D. The paired $t$ test in Graphpad software was used for the statistical analysis. The significance is indicated as follows: *$p < 0.05$; **$p < 0.01$; ***$p < 0.001$.

**Western blot analysis**. Cells were lysed on ice in SDS lysis buffer (50 mM Tris-HCl, 150 mM NaCl, 5 mM EDTA, 1% TritonX-100, 0.5% Na-Deoxychote, 1× Protein inhibitor, 1× DTT) for 30 min, shaking for 30 s every 5 min. Protein samples were resolved by SDS-PAGE (EpiZyme) and transformed to the PVDF membranes. The blots were incubated over night at 4 °C with primary antibodies against Anti-CD31 (Abcam, ab28364, 1:500), Anti VE-cadherin (Abcam, ab33168, 1:1000), Anti-NFATc1 (Abcam, ab2796, 1:2000), Anti-ISLET1 (Abcam, ab20670, 1:1000), FSP1 (Abcam, ab124805, 1:1000), anti-TBX2 (Proteintech, 22346-1-AP, 1:200), Anti-BMP4 (Abcam, ab39973, 1:1000) and Anti-GAPDH (Santa Cruz, sc-25778, 1:1000), followed by incubation with a HRP-conjugated goat anti-rabbit IgG (GtxRb-003-DHRPX, ImmunoReagents, 1:5000), a HRP-linked anti-mouse IgG (7076S, Cell Signaling Technology, 1:5000) for 1 h at room temperature. Western blotting was detected by ECL substrate (Advansta, K-12045-D20) and visualized by LAS4000 mini luminescent image analyzer (GE Healthcare Life Sciences, USA).

**Flow cytometry assay**. Cells were washed twice with DPBS (BI, China), and digested with Typsin (BI) for 1 min, folowed by wash with DPBS containing 0.5% BSA (0.5% PBSA). Next, cells were centrifuged and washed once with 0.5% PBSA. The resuspended cells in 200 μl 0.5% PBSA were passed the flow tube to obtain single cells. After that, cells were incubated with the antibodies for 30 min at room temperature. The antibodies used are Anti-NFATc1 (Abcam, ab2796); Anti-ISL1 (Abcam, ab178400); Anti-VE-cadherin (Abcam, ab33168), Anti-CD31 (R&D, FAB35679), Anti-KDR (R&D, FAB3579), Anti-CDH5 (BD Horizon, 561569), Anti-SOX9 (Proteintech, 67439-1-Ig, 1: 200), Anti-HEY1 (Proteintech, 19929-1-AP, 1: 200), Anti-JAG1 (Proteintech, 668909-1-Ig, 1: 200) or (Invitrogen, PA5-86057), Anti-NOTCH1 (Proteintech, 20687-1-AP, 1: 200), Anti-NOTCH4 (Abcam, ab225329, 1:100), Anti-P-SELECTIN (Proteintech, 60322-1-Ig, 1:200) and the isotype control antibodies: mouse IgG (R&D, C002P) and rabbit IgG (Invitrogen, 10500C). After three times washing, cells were treated with secondary

antibodies at room temperature for 15 min. Finally, cells were washed twice and resuspended with 300–500 μl 0.5% PBSA, and then analyzed by Accuri C6 flow cytometer (BD Biosciences, USA).

**Immunofluorescence**. Cells were fixed in 4% paraformaldehyde (PFA) for 15 min at room temperature, then were washed 3 times with DPBS containing 5% Triton X-100 for 10 min. Following the incubation with blocking buffer (5% normal horse serum, 0.1% Triton X-100, in PBS) at room temperature for 1 h, cells were incubated with primary antibodies at 4 °C overnight. The primary antibodies used were: ISL1 (Abcam, ab178400, 1:300), CD31 (Abcam, ab28364, 1:80), NOTCH4 (Abcam, ab225329, 1:100), DLL4 (Abcam, ab7280); VE-cadherin (Abcam, ab33168, 1:300), CDH5 (proteintech, 66804-4-lg, 1:200), NFATc1 (Abcam, ab2796, 1:100), Endomucin (Abcam, ab106100, 1:50), HEY1 (Proteintech, 19929-1-AP, 1: 200), LDB2 (Abcam, ab3627, 1:100), GATA4 (Proteintech, 19530-1-AP, 1:100), Anti-PROX1 (Proteintech, 11067-2-AP, 1:200) and NESTIN (Abcam, ab22035, 1:200). After three-times washing with PBST, the cells were incubated with secondary antibodies (1: 500 dilution in antibody buffer, Alexa Fluor-488 or -555, ThermoFisher) at room temperature for 1 h in the dark. The nuclei were stained with DAPI (D9542, Sigma, 1:1000). After washing with PBS twice, the slides were mounted with 100% glycerol on histological slides. Images were taken by a Leica SP8 laser scanning confocal microscope (Wetzlar, Germany).

Quantification of immunofluorescence staining was done by ImageJ software. When measuring the number of positive cells, 3–4 random fields per coverslip were counted. DAPI-positive cells (a total of appropriately 500 cells) were counted as the total number of cells. The proportion of cells positive for specific markers was calculated with respect to the total number of DAPI-positive cells, and the results were expressed as the mean ± s.e.m. of cells in 5–6 fields taken from 3 to 4 cultures of three independent experiments. Differences in means were statistically significant when $p < 0.05$. Significant levels are: *$p < 0.05$; **$P < 0.01$.

**Chromatin immunoprecipation (ChIP)**. ChIP experiments were performed according to the Agilent Mammalian ChIP-on-chip manual as described[63]. Briefly, $1 \times 10^8$ cells were fixed with 1% formaldehyde for 10 min at room temperature. Then the reactions were stopped by 0.125 M Glycine for 5 min with rotating. The fixed chromatin were sonicated to an average of 500–1000 bp (for ChIP-qPCR) using the S2 Covaris Sonication System (USA) according to the manual. Then Triton X-100 was added to the sonicated chromatin solutions to a final concentration of 0.1%. After centrifugation, 50 μl of supernatants were saved as input. The remainder of the chromatin solution was incubated with Dynabeads previously coupled with 10 μg ChIP grade antibodies (SMAD4, #9515, CST) overnight at 4 °C with rotation. Next day, after 7 times washing with the wash buffer, the complexes were reverse cross-linked overnight at 65 °C. DNAs were extracted by hydroxybenzene-chloroform-isoamyl alcohol and purified by a Phase Lock Gel (Tiangen, China). The ChIPed DNA were dissolved in 100 μl distilled water. Quantitative real-time PCR (qRT-PCR) was performed using a Bio-Rad qPCR instrument. The enrichment was calculated relative to the amount of input as described. All experiments were repeated at least two times. The relative gene expression levels were calculated based on the $2^{-\Delta\Delta Ct}$ method. Data were shown as means ± S.D. The paired $t$ test was used for the statistical analysis. The significance is indicated as follows: *$p < 0.05$; **$p < 0.01$; ***$p < 0.001$.

**Shear stress experiments**. The shear stress experiments were performed using Ibidi Pump System (Cat#: 10902, Ibidi). Approximately $1–2 \times 10^4/cm^2$ hPSC-derived VELs, primary VECs, and HAEC/HUVEC were plated on μ-Slide I Luer 0.4 mm (Cat#: 80176). Then, using Ibidi Pump System, steady, unidirectional laminar shear stress of 15 dynes/cm$^2$ was applied to monolayers of cells for 48 h, with static cultures serving as controls. After treatment, the slides were stained for F-actin (rhodamine phalloidin, Molecular Probes #R415, 1:400) and CD31. The samples were imaged using laser confocal microscopy. Or the cells were collected for extraction of total RNA and used for qRT-PCR analysis.

**Low-density lipoprotein uptake assay**. HPSC-derived VELs, primary VECs, and H9 ESCs (as negative control) were cultured to a confluency of 30–40%. After serum starve of the cells for 12 h, cells were incubated at a final concentration of 15 ng/ml of Alexa Fluor 594 AcLDL (Invitrogen) for 4 h at 37 °C. The cells were rinsed twice with DPBS and were fixed with 4% PFA for 10 min at room temperature. The nuclei were stained with DAPI (D9542, Sigma, 1:1000). Images were taken by a Leica SP8 laser scanning confocal microscope (Wetzlar, Germany).

**In vitro tube formation assay**. HPSC-derived VELs, HAEC, and the isolated primary VECs were cultured on a Matrigel (Corning) coated 96-well plate (Matrigel, #356234, Corning, USA) in EBM-2 medium (Lonza). Images were taken 24 h later after plating under a phase-contrast microscope. Quantification of capillary-like structures was performed by Angiogenesis Analyzer in the ImageJ, downloaded at https://imagej.nih.gov/ij/macros/toolsets/.

**Assay of cell shape and orientation**. Cell shape was evaluated by cell elongation factor (cell length along flow direction divided by cell width) and cell roundness[64].

Cell orientation angle (positive value of degrees) was measured by ImageJ. Orientation of cells related to the flow direction. Thereby, 0 degree and 90 degree alignment angle represents parallel and perpendicular, respectively. At least 50 cells were evaluated per condition across three independent replicates.

**Interaction between the de-cellularized porcine valves and the VELs**. Approximately $5 \times 10^5$ hPSC-derived VELs, the isolated primary VECs and HAEC/HUVEC were added to the apical surface of the DCV constructs and allowing attachment for 24 h in a $CO_2$ incubator. Then the medium was removed and tissue constructs were carefully turned over using sterile forceps, and cells seeding procedure was repeated on the opposite surface of the constructs. The constructs were maintained in culture for a week with medium changed daily or 3–4 weeks to study the long term interaction between the seeded hPSC-derived VELs and the de-cellularized porcine aortic valves.

**Transwell assay**. HPSC-derived VELs and the isolated VECs were washed twice with PBS. A total of $8 \times 10^4$ cells were suspended in 200 μl serum-free EBM-2 and seeded in the upper chamber of a transwell system (3422, Corning, USA). The lower chamber was filled with 600 μl EBM-2 containing supplements and growth factors. Cells were allowed to migrate for 6 h before membranes were fixed with 4% PFA for 10 min and stained with 0.5% crystal violet dyes for 2 h. The cells on the upper chambers was scraped with a cotton swab. Random fields were photographed under a microscope and cells were counted.

**TUNEL staining assay**. HPSC-derived VELs, the isolated primary VECs, and HAEC were seeded onto a 96-well plate. After washing twice with PBS, cells were treated with H2O2 (200 μM) for 24 h. Then TUNEL staining was performed using a TUNEL detection kit (Vazyme biotech Co. Ltd., China) according to the manufacturer's instructions. The microscopic areas were randomly selected and the TUNEL-positive cells were calculated by Image-Pro Plus.

**EdU incorporation analysis**. HPSC-derived VELs, the isolated primary VECs, and HAEC were seeded on the de-cellularized porcine aortic valves. The EBM-2 culture medium or our home made medium (VEGFA+ BMP4+ TGFb1) was added with 5-ethynyl-2-deoxyuridine (EdU, Ribobio, China). Cells were fixed at 72 h and 96 h after plating and processed for immunofluorescence by using the Cell-Light EdU Apollo567 In Vitro Kit (C10310-1, Ribobio, China). Images were captured and the number of EdU positive nuclei was counted manually by using Image-Pro Plus software.

**Cell viability analysis**. HPSC-derived VELs, the isolated VECs and HAEC were seeded the de-cellularized porcine aortic valves, and seeded at the density of $7 \times 10^3$ cells/well. At day 1, 3, 5, and 7 after plating, cells were rinsed with PBS and cultured in 100 μl EBM-2, followed by the addition of 20 μl MTS solution (Promega, USA). A triple number of wells were set for each time point. Cell viability was measured with a spectrophotometer at an absorbance of 490 nm after 1 h incubation.

**Statistics and reproducibility**. All experiments were repeated at least two times. Data are presented as mean values ± SEM unless otherwise stated. The paired $t$ and two-tailed unpaired $t$ test in Graphpad software were used for the statistical analysis depending on the data. Significant levels are: $*p < 0.05$; $**P < 0.01$; $***P < 0.001$.

**Reporting summary**. Further information on research design is available in the Nature Research Reporting Summary linked to this article.

## Data availability

All RNA-seq data have been deposited into the database at https://bigd.big.ac.cn/. The accession number is PRJCA002549. Uncropped western blots are provided in the Supplementary data file. Source data underlying the graphs and charts presented in the figures are available in the Supplementary Data. All other related data will be available upon reasonable request.

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

## Acknowledgements

We thank Prof. Donghui Zhang from Hubei University (Wuhan, China) for providing the PGP1 iPSCs. We thank Prof. Ning Wang from Illinois University for the comments of manuscript. This work was supported by National Key Research and Development Program of China (2016YFA0101100), National Natural Science Foundation of China (31671526) to Y. Sun, and National Natural Science Foundation of China (81930052) to N.G. Dong.

## Author contributions

L.X. Cheng, Y.Y. Zhang, and Y.C. Geng performed the hPSC VEC differentiation. L.X. Cheng also performed bioinformatics analysis; M.H. Xie, W.H. Qiao, and Y.u. Song performed the interaction of hPSC-derived VECs with DCVs; W.L. Xu, Z. Wang, L. Wang, K. Huang, N.G. Dong, and Y.H. Sun supervised the study.

## Competing interests

The authors declare no competing interests.
