## [Peer Review File · Communications Biology]

Reviewers' comments:

Reviewer #1 (Remarks to the Author):

to valve endothelial-like cells (VELs), a presumed precursor to valve endothelial cells (VECs). The authors show that the VELs express key markers for VECs and exhibit proliferative and clonogenic potential. The results are potentially novel, but questions about the extent of differentiation and function of the VELs need to be resolved.

While there are gene expression comparisons between the VELs and embryonic cardiac cells and between the VELs and VECs, good functional measures are not provided to show how similar the VELs are to VECs. The functional assays aren't convincing that the cells are similar to valvular endothelial cells. Acetylated LDL uptake is not unique to VELs or VECs. Tube formation assay results (Figure 6b) does not look like results of other assays for tube formation. In both cases, further quantification is needed along with positive and negative controls.

These conclusions would be stronger if the VECs are included and assays performed provide unique VEC behavior. Unique responses of VECs include alignment perpendicular to flow (Arteriosclerosis, Thrombosis, and Vascular Biology. 2004;24:1429-1434) and expression of VCAM-1 and ICAM-1 (Arteriosclerosis, Thrombosis, and Vascular Biology. 2009;29:254-260) or effect of shear stress on NOTCH1 and repression of tissue calcification (J. Mol. Cell Cardiol. 2015, 84:13-23).

Figure 3 is confusing. The title refers to VECs but the text indicates that the embryonic cardiac cells are analyzed and compared with the VELs. Figures 3a, b, c are embryonic cardiac tissue. Further, the VELs are not presented in these graphs, so one cannot easily assess the claim that the VELs resemble embryonic tissue.

Specific Comments

1. p 11 Provide a citation(s) for the following statement: "It is widely accepted that hPSC-derived derivatives in vitro resemble embryonic type cells more than adult." Is it a sequential or branching process?
2. Figure 2a doesn't provide much useful information. The diagram would be better if it included the transition of CPC to ECC as well as the VELs and VECs indicating key markers for each cell type.
3. Figure 2c and associated text. The pattern of gene expression is more variable than indicated. Results for NPR3, ENG, and NOTCh1 were not shown. DLL4 shows a delayed induction. Perhaps the only trend is that the VEC genes have a common feature that they reach a maximum later than the ECC genes.
4. Figure 3b. Explain the meaning of the x and y axes and their relationship to the axes in Figure 3a.
5. Figure 3 Legend. Provide citation for stochastic neighbor embedding method.
6. Figure 3f legend. Define H9.
7. The terms VEL and VEC-like are not clearly defined and the distinction between the two, if any, needs clarification.
8. Flow of text on p21 is choppy. Suggest focusing on differentiation and early passage first, then discuss late passage and loss of VEL phenotype. Provide additional evidence that phenotype is lost by looking at VEL markers at different times after differentiation.
9. Supplementary Figure 6. Evidence for dedifferentiation is limited. Immuno-fluorescent staining would strengthen the conclusion.
10. Figure 8 - Do VELs form a confluent monolayer on porcine heart valves? VECs are needed for comparison to VELs.
11. Figure 8b and 8d and associated text on p 24.- Data are needed to support statement that VELs are more proliferative than VECs.
12. Figure 8f - Indicate in legend VEC marker used to identify VELs.

Reviewer #2 (Remarks to the Author):

The manuscript describes the generation and characterization of cardiac valve endothelial-like cells from human pluripotent stem cells.

The authors describe extensively and in a step-wise manner why and which components are used in the proposed differentiation protocol to generate Cardiac Valvular Endothelial-Like cells (VELs) from iPSCs.

After doing so the authors report on an equally extensive comparison of these aforementioned cells to native Valvular Endothelial Cells. Various analysis are demonstrated side by side.

In the end the iPSC-derived VELs are seeded onto a porcine decellularized valve to demonstrate cell spreading and clonogenicity.

In general terms the authors report on a very well-documented iPSC differentiation protocol rendering VELs (and potentially also VICs, valvular interstitial cells) for further study. In itself this may be of interest to the field of regenerative medicine, especially to those interested in valvular recellularization. Other than this the depth of the work appears to be limited.

1) The data in the manuscript are application-focussed and are limited in hypothesis-driven research.

2) The first deliverable of the manuscript as indicated in the introduction "a platform for mechanistic study of valvulogenesis" is not met in a satisfactory manner.

a) The molecular mechanisms underlying the differentiation protocol are indicated by big-data analysis. Again in itself well conducted, but lacking in e.g. gene-edited interventional studies that underpin these proposed mechanisms.

b) The physiological mechanistics that may be associated with valvulogenesis are not addressed at all. Neither descriptive nor experimental.

3) The second deliverable of the manuscript as indicated in the introduction ".. a starting material for the construction of valve organoids..." is not supported by the data provided. Other than that the VELs grow on a decellularized matrix there is no construction of valve leaflets, no hypothesis driven study into the effect of such constructs.

4) The variation and mutual corroboration of the techniques and resulting data presented is the strong point of this manuscript. The experiments appear relevant and the data are sound and well presented.

a) The description and explanation of the two-stage differentiation protocol reads as all-encompassing narrative on the components used. The authors are praised for this level of detail.

b) The characterization of the iPSC-derived VELs is extensive. The authors used a comprehensive set of protein / gene / surface marker expression analyses to follow cell fate. These experiments are well executed, are mutually supportive and are well presented. [Fig 1, Suppl Fig1, Fig2, Suppl Fig2]

c) The VELs are compared to the VECs on a single cell gene expression level rendering a wealth of expression data [Fig 3, Suppl. Fig 3] and global transcriptome comparison [Fig 4, suppl Fig 4]. These experiments are well executed and are well presented.

d) The involvement of the known association between TGFb1/BMP-4 signaling via NFATc1 is the main mechanistic / hypothesis driven set of experiments [Fig 5, Suppl Fig5]. These experiments are well executed, are mutually supportive and are well presented.

e) The characterization of VEL cell function [Fig 6, Suppl Fig6] and
f) the involvement of endo MT in the generation of VICs are well executed and well presented. [Fig7].
The authors may want to correct the type-O in Fig 7e: "membarne" .
g) Cell seeding of VELs on decellularized porcine valves is also well executed and well presented [Fig8,
Suppl Fig8]

5) The manuscript would benefit from experiments taking the knowledge on generating the VELs and also the VICs to a higher level: Is there a functional improvement of valve function in a transplantation model. E.g. would VEL coverage of a decellularized valve reduce thrombogenicity / inflammation of a valvular graft. E.g. Would the application of VICs improve the mechanical properties or remodeling of a valvular graft.?

Reviewer #3 (Remarks to the Author):

This manuscript by Cheng et al. aims to develop a chemically defined in vitro system to produce hPSCs-derived valve endothelial-like cells (VELs) and to recapitulate the developmental features of human heart valvulogenesis, which involves multipotent cardiac progenitor cells (CPCs) generation, endocardial cushion cell (ECCs) specialization, and endocardial to mesenchymal transition. This topic is of great interest to the field, and there is an impressive amount of data in the manuscript; however the data demonstrating the resemblance of hPSC derived endothelial cells and primary VECs is not very convincing. Unfortunately, the manuscript is not well-organized, is missing some key references, lacks data analysis details, and contains some over-statements and typos. The experimental procedures need further clarification, and the figure legends should have been carefully proofread prior to submitting the manuscript. A well-organized version of this manuscript could be an important contribution to the field.

Some examples include:

- 1) Add missing references: Glaser et al. ,2011; Hoffman et al., 2011; Wu et al. 2013; Wang et al. 2020; Mandal et al. 2016; Timmerman et al. 2017; Vincent and Buckingham 2010; Lee et al. 1994; Lopez-Sanchez et al. 2015; Singh et al. 2011;
- 2) Correct the spelling for the following author names: Adersen et al, 2018; Muhamud et al, 2019; El-Ras et al. 2017; Hoogaars et al. 2007; Zavadil et al. 2004;
- 3) Supplement Fig.2 legend: The description does not match the data.
- 4) Specifics of CHIR99021 application is missing (the dose and the result comparing to WNT-3a);
- 5) What is the cell source for HFF and hVICs?
- 6) Fig. 1: indicate the treatment condition for 3 days CPC: WNT3a+BMP for 3 days or Day1 WNT3a+BMP, D2 and D3 BMP+bFGF. Also, provide the specifics of all the t-test performed throughout the paper. Unpaired t-test?
- 7) Fig.1e: Is the internal control (GAPDH) derived from the same SDS-PAGE gel with the other three proteins (NKX2.5, ISL1, and KDR)? add molecular weight for GAPDH.
- 8) Fig.2h: Is the internal control (GAPDH) derived from the same SDS-PAGE gel with the other 7 proteins (NFATc1, NOTCH1, TBX2, HEY1, HEY2, DLL4, NOTCH3)? add molecular weight for GAPDH.
- 9) Fig. 3a and 3b: How many cells are plotted in the panels? There are 13 time-points and probably less than 100-200 dots in the plot. Assuming each dot is a single cell, did the authors get ~10 cells per time-point? Why are there so few cells? Why did they pick these ~10 cells per condition? Also, the trajectory is not as clear as the authors claimed in the main text. There are 13 week cells positioned after to 24-25 week cells.
- 10) Fig. 3g: The authors claim there are 14 cell clusters in the data. Do each of these clusters have unique gene identifiers? What is the effect of technical differences (coverage etc.) in clustering? How was the cell clustering done? What were the parameters used for the analysis?
- 11) Fig. 3f: Was this heatmap scaled by row? Otherwise, why are the RPKM values so low? It appears

to us that those are Z values possibly, not RPKM. We noticed the same issue in the other heatmaps as well. All heatmaps show Log₂RPKM, is that really the case?

12) Fig. 4b: The global profiles of VELs and primary VECs do not resemble to each other. Then, the authors pick a handful of genes to show some similarities in 4e but the data in 4b is concerning.

13) Fig. 8: The data in this figure should be supported with more stainings and comparisons (quantifications from both conditions) between hPSC-derived VELs and primary endothelial cells.

14) Discussion: Cardiac valves include atrio-ventricular (AV) and semilunar valves. Although previous studies have identified similar molecular and cellular processes in both valve systems, there is also strong evidence that cardiac neural crest cells (cNCCs) contribute to the valvulogenesis (Kirby 1983, Camenisch 2002, Delton 2003, Geoge 2020Henderson 2020). The manuscript should mention the limitations of the current mesoderm-based system.

Reviewers' comments:

Reviewer #1 (Remarks to the Author):

to valve endothelial-like cells (VELs), a presumed precursor to valve endothelial cells (VECs). The authors show that the VELs express key markers for VECs and exhibit proliferative and clonogenic potential. The results are potentially novel, but questions about the extent of differentiation and function of the VELs need to be resolved.

Response: we appreciate the reviewer for the encouraging and nourishing comments, which helped us greatly improved the manuscript. In the revised manuscript, we have added new data to address the extent of differentiation and lack of functional assay of VELs.

While there are gene expression comparisons between the VELs and embryonic cardiac cells and between the VELs and VECs, **good functional measures** are not provided to show how similar the VELs are to VECs. The functional assays aren't convincing that the cells are similar to valvular endothelial cells. Acetylated LDL uptake is not unique to VELs or VECs. Tube formation assay results (Figure 6b) does not look like results of other assays for tube formation. In both cases, further quantification is needed along with positive and negative controls.

Response: we thank the reviewer for the comments. In the revised manuscript, we have added new data for the functional assays for hPSC-derived VELs, primary VECs (used as positive control) and HAEC (used as negative control)(new Figures 6d-f; Figures 8b-e). Tube formation assay was repeated with controls (Figure 6b), and additional functional assay was also added as shown. Please find the text information at

Pages 21-22, and 24-25.

These conclusions would be stronger if the VECs are included and assays performed provide unique VEC behavior. Unique responses of VECs include alignment perpendicular to flow (Arteriosclerosis, Thrombosis, and Vascular Biology. 2004;24:1429 - 1434) and expression of VCAM-1 and ICAM-1 (Arteriosclerosis, Thrombosis, and Vascular Biology. 2009;29:254 - 260) or effect of shear stress on NOTCH1 and repression of tissue calcification (b).

Response: We thank the reviewer for these detailed experiments that are essential. In the revised manuscript, we have performed new experiments that are unique to VECs, including examining the cell alignment to the shear stress for hPSC-derived VELs, VECs and HAEC, and expression of *Vcam1/Icam1* as well as the Notch-related genes in response to the shear stress.

We found that expression of *Vcam1/Icam1* as well as the Notch-related genes was comparable between VELs and VECs, in response to the shear stress. However, we did notice that hHPSC-derived VELs partially resembled the primary VECs in cell alignment (the later is more perpendicular orientation to flow direction). We think that this result might be due to: 1) we have set the fluid flow at 15 dynes/cm², while Butcher et al. used 20 dynes/cm²; and 2) the hPSC-derived VELs generated by current method remain immature, although they resemble the primary VECs in terms of gene expression.

Please find the new data in the new Figures 6e-f and text at Pages 21-22.

Figure 3 is confusing. The title refers to VECs but the text indicates that the embryonic cardiac cells are analyzed and compared with the

VELs. Figures 3a, b, c are embryonic cardiac tissue. Further, the VELs are not presented in these graphs, so one cannot easily assess the claim that the VELs resemble embryonic tissue.

Response: We thank the reviewer for pointing this out. In the revised manuscript, we have extensively revised the Figure 3 and the corresponding text content. Now the figure 3 is used to focus on demonstrating that hPSC-derived VELs resemble the embryonic VECs, and that our hPSC VEL differentiation recapitulates the development of embryonic VECs. Please find the more information at Figure 3 and the text at Pages 12-15.

At the same time, we have combined the scRNA data in previous Figure 3 with Figure 4 to compare the similarity between the VELs and VECs from different ages (please see Figure 4 and the text at pages 17-18).

In sum, now the new Figure 3 is used to show that the VELs resemble embryonic tissue, and the new Figure 4 is used to show that the VELs partially resemble VECs isolated from adults.

Specific Comments

1. p 11 Provide a citation(s) for the following statement: “It is widely accepted that hPSC-derived derivatives in vitro resemble embryonic type cells more than adult.” Is it a sequential or branching process?

Response: We thank the reviewer for the comment. To our knowledge, this point of review is widely appreciated in the stem cell community (as in vitro PSC-derived cells are often immature, and therefore more like the embryonic type cells). However, to avoid any confusion, we deleted this sentence instead in the revised manuscript (so no citation).

2. Figure 2a doesn't provide much useful information. The diagram would be better if it included the transition of CPC to ECC as well as the VELs and VECs indicating key markers for each cell type.

Response: We thank the reviewer for the advice. We have edited the figure 2a accordingly.

3. Figure 2c and associated text. The pattern of gene expression is more variable than indicated. Results for NPR3, ENG, and NOTCH1 were not shown. DLL4 shows a delayed induction. Perhaps the only trend is that the VEC genes have a common feature that they reach a maximum later than the ECC genes.

Response: We thank the reviewer for the comments. The previous figure was not well-organized and the data was not well-presented. In the past few months, we have repeated the PSC differentiation and repeated the qRT-PCR assay for genes including *DLL4*, *NPR3* and *NOTCH1*. As you can see in the new Figure 2c, the expression pattern for both ECC and VEC genes is more general (consistent) than before. For instance, ECC markers all peak at 4 day post-treatment, and VEC markers peak at day 6 post-treatment.

4. Figure 3b. Explain the meaning of the x and y axes and their relationship to the axes in Figure 3a.

Response: Figure 3a is the tSNE assay showing there are 4 EC cells based on the re-analysis of reported scRNA data (Cui et al., 2019). Figure 3b was the pseudotime trajectory results used to show the developmental progression of embryonic ECs. We only extracted the endocardial and valvular EC populations for the assay (because VECs originate from endocardial cells, but not from vascular EC lineage).

To make the Figure 3b more self-explaining and more meaningful, we have added two additional panels (please see the new Figure 3b). The x (component 1) and y (component 2) axes were just used to show the data by the R software. For more detail information of how we analyzed the data, please refer to the method part of “Single cell RNA-sequencing analysis” (Pages 36-37).

5. Figure 3 Legend. Provide citation for stochastic neighbor embedding method.

The t-distributed stochastic neighbor embedding (t-SNE) is the common calculating method used for the assay of scRNA data using seurat package. This has been described in detail by Cui et al., 2019, and we have cited this paper (please see the legend in Figure 3 and the text at pages 11-12). You may please also see the method part of “Single cell RNA-sequencing analysis” at Pages 36-37.

6. Figure 3f legend. Define H9.

This was defined, now in Figure 3e legend and in the text (page 15).

7. The terms VEL and VEC-like are not clearly defined and the distinction between the two, if any, needs clarification.

Thanks for pointing this out. VEC-like is the same as VEL. To avoid any confusion, I have clarified it and have made it clear when the VEC-like was first appeared in the text (page 6 line 1).

8. Flow of text on p21 is choppy. Suggest focusing on differentiation and early passage first, then discuss late passage and loss of VEL phenotype. Provide additional evidence that phenotype is lost by looking at VEL markers at different times after differentiation.

We thank the reviewer for the comments. We have edited the text accordingly (Figures S6a-b; page 20).

9. Supplementary Figure 6. Evidence for dedifferentiation is limited. Immuno-fluorescent staining would strengthen the conclusion.

Immuno-fluorescent staining data have been added in the revised Figure S6, in which we used CD31 for EC marker and SMA for differentiating marker.

10. Figure 8 - Do VELs form a confluent monolayer on porcine heart valves? VECs are needed for comparison to VELs.

Yes, in our hand, VELs attach to the DCVs and grow as monolayer after seeding. In the revised manuscript, VECs and HAEC are added for comparison.

11. Figure 8b and 8d and associated text on p 24. - Data are needed to support statement that VELs are more proliferative than VECs.

We thank the reviewer for the comments. In the new Figure 8, we have repeated both TUNEL and EdU assay. We observed that VELs are more proliferative revealed by EdU staining than VECs at least two time points (72 and 96h) after seeding (Pages 25-26).

12. Figure 8f - Indicate in legend VEC marker used to identify VELs.

Previous description was not accurate, as the experiment was HE staining not IHC performed by our collaborator. I apologize for this.

We have repeated at least two times the HE after seeding cells on the DCVs. We found that VELs are prone to attach better than HAEC and VECs, as shown in the new Figure 8 (now in Figure 8e).

Reviewer #2 (Remarks to the Author):

The manuscript describes the generation and characterization of cardiac valve endothelial-like cells from human pluripotent stem cells. The authors describe extensively and in a step-wise manner why and which components are used in the proposed differentiation protocol to generate Cardiac Valvular Endothelial-Like cells (VELs) from iPSCs. After doing so the authors report on an equally extensive comparison of these aforementioned cells to native Valvular Endothelial Cells. Various analysis are demonstrated side by side. In the end the iPSC-derived VELs are seeded onto a porcine decellularized valve to demonstrate cell spreading and clonogenicity.

In general terms the authors report on a very well-documented iPSC differentiation protocol rendering VELs (and potentially also VICs, valvular interstitial cells) for further study. In itself this may be of interest to the field of regenerative medicine, especially to those interested in valvular recellularization. Other than this **the depth of the work appears to be limited.**

1) The data in the manuscript are application-focused and are limited in hypothesis-driven research.

Response: we appreciate the reviewer for the important comments. This work is indeed application oriented and in particular primarily trying to generate VEC-like and VIC-like cells from PSCs that are close to the genuine valvular cells.

The current manuscript is indeed lacking of mechanistic studies. Followed by this work, we planned to do some hypothesis-driven research and investigate the mechanisms underlying VEC formation, for instance, how the stepwise signaling activation promotes VEC formation or why

NFATc1/ATF3/KLFs might be required for VEC induction or formation by generating gene knockout PSCs.

2) The first deliverable of the manuscript as indicated in the introduction "a platform for mechanistic study of valvulogenesis" is not met in a satisfactory manner.

a) The molecular mechanisms underlying the differentiation protocol are indicated by big-data analysis. Again in itself well conducted, but lacking in e.g. gene-edited interventional studies that underpin these proposed mechanisms.

Response: we appreciate the reviewer for pointing out this. We meant to express that "PSC differentiation to VECs provides a platform for future mechanistic study of valvulogenesis".

Because of lack of gene-edited based genetic analysis, our previous statement "a platform for mechanistic study of valvulogenesis" was overstated in its current form. As you suggested, we have removed this sentence or used more appropriate words in the introduction (Page 6).

b) The physiological mechanistics that may be associated with valvulogenesis are not addressed at all. Neither descriptive nor experimental.

Response: we appreciate the reviewer for the important comment. In the revised manuscript, we have added some new experimental data, for instance, the gene expression of VELs in response to flow shear stress and the cell alignment in response to fluid flow, which partially mimic the physiological conditions of primary VECs (which are under constant hemodynamic stimulus). Meanwhile, we have add some discussion for the physiological mechanistics (pages 30-31).

3) The second deliverable of the manuscript as indicated in the introduction "... a starting material for the construction of valve organoids..." is not supported by the data provided. Other than that the VELs grow on a decellularized matrix there is no construction of valve leaflets, no hypothesis driven study into the effect of such constructs.

Response: we appreciate the reviewer for the comments. In the revised manuscript, we have added some new data describing the interaction between HPSC-derived VELs and de-cellularized porcine heart valves, in particular we showed that the re-endothelialization occurred on the DCVs (as you know, the endothelialization is the important step for making the hPSC-based valve organoids). We think that these experiments are the important step for future construction of valve leaflets. Next, we will investigate whether endoEMT can occur after long term culture or construct valve leaflets using our PSC-derived valvular cells and then transplant the valve leaflets into the animal model to evaluate its performance in physiological conditions. We hypothesize that the PSC derived-organoids or valve leaflets perform better than the current mechanical and bioprosthetic valves in functional assays, say by increasing the durability with self-growth and self-renewal capacity while reducing thrombogenicity/inflammation.

As the reviewer pointed out, the sentence "a starting material for

the construction of valve organoids” was over-stated in the current manuscript, and therefore has been revised accordingly (page 6).

4) The variation and mutual corroboration of the techniques and resulting data presented is the strong point of this manuscript. The experiments appear relevant and the data are sound and well presented.

a) The description and explanation of the two-stage differentiation protocol reads as all-encompassing narrative on the components used. The authors are praised for this level of detail.

Response: we thank the reviewer for encouraging us in this aspect.

b) The characterization of the iPSC-derived VELs is extensive. The authors used a comprehensive set of protein / gene / surface marker expression analyses to follow cell fate. These experiments are well executed, are mutually supportive and are well presented. [Fig 1, Suppl Fig1, Fig2, Suppl Fig2]

Response: we thank the reviewer for the comment.

c) The VELs are compared to the VECs on a single cell gene expression level rendering a wealth of expression data [Fig 3, Suppl. Fig 3] and global transcriptome comparison [Fig 4, suppl Fig 4] .These experiments are well executed and are well presented.

Response: We have added additional panel for new Figure 3b, hoping that it is more self-explaining and clearer.

d) The involvement of the known association between TGFb1/BMP-4 signaling via NFATc1 is the main mechanistic / hypothesis driven set of experiments [Fig 5, Suppl Fig5]. These experiments are well executed, are mutually supportive and are well presented.

Response: More genetic experiments will be performed, following this work.

e) The characterization of VEL cell function [Fig 6, Suppl Fig6] and

f) the involvement of endo MT in the generation of VICs are well executed and well presented. [Fig7]. The authors may want to correct the type-0 in Fig 7e: "membarne" .

Response: we have corrected the typo (figure 7e). Thanks

g) Cell seeding of VELs on decellularized porcine valves is also well executed and well presented [Fig8, Suppl Fig8]

Response: we have improved a little in the revised manuscript as suggested by other reviewers, with some positive and negative controls.

5) The manuscript would benefit from experiments taking the knowledge on generating the VELs and also the VICs to a higher level: Is there a functional improvement of valve function in a transplantation model. E.g. would VEL coverage of a decellularized valve reduce thrombogenicity / inflammation of a valvular graft. E.g. Would the application of VICs improve the mechanical properties or remodeling of a valvular graft.?

Response: we thank the reviewer for the inspiring comments. Actually, these are the next-step target of our study: build up new-type tissue-engineered scaffolds (organoids) implanted with PSC-derived VELs and VICs, and transplant them into big animal model to investigate whether the PSC-based valve organoids would outperform the DCVs or the current tissue-engineered scaffolds, by reducing the thrombogenicity/inflammation while improving the mechanical properties and duration as well as the capacity of self-growing/-renewal.

As discussed above, the current manuscript is primarily focused on obtaining the seed cells: genuine valvular cells (VECs and VICs) from PSCs. In fact, we are doing some functional assays and animal experiments in the lab. Our group is composed of excellent physicians from hospital, stem cell people from the institute, material and tissue-engineering experts from universities, and we wish we deepen our

collaboration and move forward to pushing our work to the next level: fabricating tissue-engineered valve leaflets implanted with PSC-derived valvular cells with better performance.

Reviewer #3 (Remarks to the Author):

This manuscript by Cheng et al. aims to develop a chemically defined in vitro system to produce hPSCs-derived valve endothelial-like cells (VELs) and to recapitulate the developmental features of human heart valvulogenesis, which involves multipotent cardiac progenitor cells (CPCs) generation, endocardial cushion cell (ECCs) specialization, and endocardial to mesenchymal transition. This topic is of great interest to the field, and there is an impressive amount of data in the manuscript; however **the data demonstrating the resemblance of hPSC derived endothelial cells and primary VECs is not very convincing**. Unfortunately, the manuscript is not well-organized, is missing some key references, **lacks data analysis details**, and contains some over-statements and typos. The experimental procedures need further clarification, and the figure legends should have been carefully proofread prior to submitting the manuscript. A well-organized version of this manuscript could be an important contribution to the field.

Response: we thank the reviewer for the critical, detailed and important comments. 1) for the comment “the data demonstrating the resemblance of hPSC derived endothelial cells and primary VECs is not very convincing”. In the revised manuscript, we have added some new data (primarily functional comparison data with positive and negative controls; in new figures 6e-f) and repeated key experiments (figure 2c; 3b), and have revised the manuscript accordingly (page 21-22; 25-26), trying to better demonstrate that VELs resemble the primary VECs.

2) “the manuscript is not well-organized, is missing some key references, lacks data analysis details, and contains some over-statements and typos”. We apologize for all this and have revised the manuscript with appropriate references, more detailed data analysis and deleted the over-statements and typos. 3) “The experimental procedures need further clarification, and the figure legends should have been carefully proofread”. In the revised manuscript, we have added more specific information to clarify the experimental procedures, including specifics of using CHIR (page 34); describing how we performed the scRNA analysis (pages 36-37); and adding controls for characterizing PSC-derived VELs. Also, we carefully proofread the manuscript before submitting to the journal.

Some examples include:

1) Add missing references: Glaser et al. ,2011; Hoffman et al., 2011; Wu et al. 2013; Wang et al. 2020; Mandal et al. 2016; Timmerman et al. 2017; Vincent and Buckingham 2010; Lee et al. 1994; Lopez-Sanchez et al. 2015; Singh et al. 2011;

Response: we thank the reviewer for pointing out the missing references. These are now added.

2) Correct the spelling for the following author names: Adsersen et al, 2018; Muhamud et al, 2019; El-Ras et al. 2017; Hoogaars et al. 2007; Zavadil et al. 2004;

Response: we apologize for these errors. Now the author names should be all right.

3) Supplement Fig.2 legend: The description does not match the data.

Response: We have double checked the legend, and now this is consistent.

4) Specifics of CHIR99021 application is missing (the dose and the result comparing to WNT-3a);

Response: Now this information is added. Please see pages 7 and 34.

5) What is the cell source for HFF and hVICs?

HFF is the human foreskin fibroblast, and hVICs stands for human valvular interstitial cells. The information was previously indicated in the text (now Page 15).

6) Fig. 1: indicate the treatment condition for 3 days CPC: WNT3a+BMP for 3 days or Day1 WNT3a+BMP, D2 and D3 BMP+bFGF. Also, provide the specifics of all the t-test performed throughout the paper. Unpaired t-test?

Response: we have dropped the previous figure 1g (as it is clearly stated the text) to make the manuscript as concise as possible. We have added the information for the t-test: the paired t-test throughout the paper.

7) Fig. 1e: Is the internal control (GAPDH) derived from the same SDS-PAGE gel with the other three proteins (NKX2.5, ISL1, and KDR)? add molecular weight for GAPDH.

Yes, it is same internal control. The molecular weight for GAPDH is added.

8) Fig. 2h: Is the internal control (GAPDH) derived from the same SDS-PAGE gel with the other 7 proteins (NFATc1, NOTCH1, TBX2, HEY1, HEY2, DLL4, NOTCH3)? add molecular weight for GAPDH.

Response: the WB experiments have been done a few times, each time with the internal control. In the revised manuscript, we have corrected this with appropriate controls (Figure 2h).

9) Fig. 3a and 3b: How many cells are plotted in the panels? There are 13 time-points and probably less than 100-200 dots in the plot. Assuming each dot is a single cell, did the authors get ~10 cells per time-point? Why are there so few cells? Why did they pick these ~10 cells per condition? Also, the trajectory is not as clear as the authors

claimed in the main text. There are 13 week cells positioned after to 24–25 week cells.

Response: We thank the reviewer for the detailed and important questions. As you may know, the collecting of human embryonic cardiac tissues is pretty difficult (we planned to do this, but failed to do so). The original scRNA data was generated by Tang' s group (Cui et al., 2019), and we re-analyzed their data for our purpose. Cui et al. have obtained a total of 3842 cardiac cells from 5W to 25W of gestation (Cui et al., 2019). Of which, the number of all cardiac ECs (including vascular ECs and valvular ECs) is 595. When we analyzing EC cells, we found that the number of endocardium and VECs is 344 and 67, respectively. So as you have pointed out, the cell number is indeed small. Nevertheless, to our knowledge, this is by far the best scRNA data that we could consult for embryonic cardiac ECs.

For the concern “the trajectory is not as clear as the authors claimed in the main text”. We have added more panels in Figure 3b, trying to make it clearer and more self-explaining. At the same time, we have edited the text accordingly. Please see the new Figure 3b and text at page 12.

10) Fig. 3g: The authors claim there are 14 cell clusters in the data. Do each of these clusters have unique gene identifiers? What is the effect of technical differences (coverage etc.) in clustering? How was the cell clustering done? What were the parameters used for the analysis?

Response: when this manuscript was under revise, our collaborators have published the analysis of scRNA data where they described the detailed information of how 14 cell clusters were identified. Please may refer

to: Xu et al., 2020. Cell-Type Transcriptome Atlas of Human Aortic Valves Reveal Cell Heterogeneity and Endothelial to Mesenchymal Transition Involved in Calcific Aortic Valve Disease. *Arteriosclerosis, Thrombosis, and Vascular Biology*. 40:2910-2921. <https://www.ahajournals.org/doi/10.1161/ATVBAHA.120.314789>.

Briefly, we performed principal component analysis on gene expression variability across all 31043 cells and then classified the cells into cell-type groups using graph-based clustering of the informative principle components (n=10). To identify the subtypes, we matched the top 100 expressed genes (fold change) in each cluster with the known markers.

In this manuscript, we only extracted the VEC clusters (total 746 cells) from the original scRNA data from 40-year old, and re-analyzed them. Briefly, using Seurat R package, we performed clustering analysis and obtained 2000 variable genes by which we used to perform PCA analysis. By tSNE in the Seurat package, we identified important PCs, and selected 15 PCs for UMAP analysis. Next, using FindClusters, we determined the two subclusters. Based on the expression of side-specific valvular EC genes such as VCAM1 and MGST1 (which are more highly expressed in aortic side than in ventricular side) (Simmons et al., 2005), we confirmed that there were two VEC subclusters. Please see the detail at pages 17-18; and method part for scRNA analysis at

pages 36–37.

11) Fig. 3f: Was this heatmap scaled by row? Otherwise, why are the RPKM values so low? It appears to us that those are Z values possibly, not RPKM. We noticed the same issue in the other heatmaps as well. All heatmaps show Log2RPKM, is that really the case?

Response: Yes, all heatmaps were scaled by row and then by log2RPKM. The Heatmap analysis was performed by HeatMap tools of TBtools by my graduate student LX Cheng who performed the bioinformatics analysis in this work. In that tool, we chose the commands: scaled by row, scaled by log, cluster rows and cluster cols.

12) Fig. 4b: The global profiles of VELs and primary VECs do not resemble to each other. Then, the authors pick a handful of genes to show some similarities in 4e but the data in 4b is concerning.

Response: The Figure 4b was based on the expression of all genes (>20000), scaled by row by pheatmap package. We wished to use this figure to show the point that there was some similarity between VELs and VECs in gene expression profile. However, with more than 20000 genes, it was difficult to get a global “very well” profile. So, at your comment, we decided to drop this panel out of the manuscript.

In the new manuscript, we keep the Figure 4e (now Figure 4d) which was generated by using the top 100 genes that are highly expressed in both the primary VECs and hPSC-derived VEL. Supplementary Figures 4b-c are also used to demonstrate this point.

13) Fig. 8: The data in this figure should be supported with more

stainings and comparisons (quantifications from both conditions) between hPSC-derived VELs and primary endothelial cells.

Response: We thank the reviewer for the comment. And we have added some new data (figures 8b-e; pages 25-26), along with quantifications.

14) Discussion: Cardiac valves include atrio-ventricular (AV) and semilunar valves. Although previous studies have identified similar molecular and cellular processes in both valve systems, there is also strong evidence that cardiac neural crest cells (cNCCs) contribute to the valvulogenesis (Kirby 1983, Camenisch 2002, Delton 2003, Geoge 2020Henderson 2020). The manuscript should mention the limitations of the current mesoderm-based system.

Response: we thank the reviewer for reminding us this important point. We have added some discussion for the contribution of cNCCs to the valvular cell formation. Please see page 27. Inspired by the reviewer, in the future, we will consider the contribution of cNCCs to the valvular cells to faithfully recapitulate the valvulogenesis.

Reviewers' comments:

Reviewer #1 (Remarks to the Author):

The study provides a detailed characterization of the differentiation of iPSCs to valve endothelial cells and their function. The revised manuscript is much improved and most of the issues raised have been addressed. The addition of the flow studies is very helpful. However, in Figure 6, functional comparisons should include primary cells along with quantification of flow response and capillary network formation. Only limited information is provided for the flow response.

Specific Comments

Figure Legends. Indicate which panels in the figure the following statement refers: "Shown are representative data."

Figure 6b. Quantify capillary-like structures in terms of branches and tube length.

Figure 6e. Quantify cell shape and orientation.

The meaning of this statement is unclear "A subset of hPSC-derived VELs partially resembled the VECs, exhibiting an angle of around 60° to the direction of flow." A histogram of the cell orientations would be helpful in understanding the degree to which the iPSC-derived cells resemble primary valve cells after exposure to flow.

Reviewer #2 (Remarks to the Author):

Dear Authors,

I thank you for your effort in upgrading the manuscript by adding new experiments and clarifying the storyline.

In addition, you also improved your work by augmenting and streamlining the figure and text.

Reviewer #3 (Remarks to the Author):

In this updated manuscript, based on the reviewers' suggestion/commons, the authors added the new designed experiments (i.e. importantly functional assay for generated VELs), reorganized the result data, and corrected most errors that existed in previous submitted manuscript.

This application-directed study focused on setting up a simple unsorting two-step human VEC-like cell differentiation model from hPSCs in the context of application of defined chemical stimulators and xeno-free medium. Accompanying with other new published protocols (i.e. Neri T et al. Nature Comm. 2019; Mikryukov AA et al. Cell Stem Cell 2021), this study did provide an optional strategy for in vitro generation of reliable source of hPSC-derived human valvular endothelial cells (VECs) (even though in the current non-mature state). The success of human cell model of valvulogenesis in dish should benefit the efforts in future performance of valve tissue engineering, in vitro valve disease modeling as well as drug screening.

Some minor modification may be suggested:

- (1) Add reference paper (Mikryukov AA et al. Cell Stem Cell, 2021) into updated manuscript;
- (2) Adjust the fig legend title for Fig.3 for the reason that Fig.3e is out of the range of "Primary VECs";
- (3) Change the color assignment in Fig Suppl.3c (not using four colors to label "DLL4", "WNT2" and two curves in one picture);
- (4) Page 2 of Result: KX2.5low should be "NKX2.5low".